# NETWORKED INEQUALITY: PREFERENTIAL ATTACHMENT BIAS IN GRAPH NEURAL NETWORK LINK PREDICTION

## ABSTRACT

Graph neural network (GNN) link prediction is increasingly deployed in citation, collaboration, and online social networks to recommend academic literature, collaborators, and friends. While prior research has investigated the dyadic fairness of GNN link prediction, the within-group fairness and "rich get richer" dynamics of link prediction remain underexplored. However, these aspects have significant consequences for degree and power imbalances in networks. In this paper, we shed light on how degree bias in networks affects Graph Convolutional Network (GCN) link prediction. In particular, we theoretically uncover that GCNs with a symmetric normalized graph filter have a within-group preferential attachment bias. We validate our theoretical analysis on real-world citation, collaboration, and online social networks. We further bridge GCN's preferential attachment bias with unfairness in link prediction and propose a new within-group fairness metric. This metric quantifies disparities in link prediction scores between social groups, towards combating the amplification of degree and power disparities. Finally, we propose a simple training-time strategy to alleviate within-group unfairness, and we show that it is effective on citation, online social, and credit networks.

## 1 INTRODUCTION

Link prediction (LP) using GNNs is increasingly leveraged to recommend friends in social networks (Fan et al., 2019; Sankar et al., 2021), as well as by scholarly tools to recommend academic literature in citation networks (Xie et al., 2021). In recent years, graph learning researchers have raised concerns with the unfairness of GNN LP (Li et al., 2021; Current et al., 2022; Li et al., 2022). This unfairness is often attributed to graph structure, including the stratification of social groups; for example, online networks are usually segregated by ethnicity (Hofstra et al., 2017). However, most fair GNN LP research has focused on dyadic fairness, i.e., satisfying some notion of parity between inter-group and intra-group link predictions. This formulation neglects: 1) LP dynamics within social groups (Kasy & Abebe, 2021); and 2) the "rich get richer" effect, i.e., the prediction of links at a higher rate between high-degree nodes (Barabási & Albert, 1999). In the context of friend recommendation systems, the "rich get richer" effect can increase the number of links formed with high-degree individuals, which boosts their influence on other individuals in the network, and consequently their power (Bashardoust et al., 2022).

In this paper, we shed light on how degree bias in networks affects GCN LP (Kipf & Welling, 2016). We focus on GCNs with symmetric and random walk normalized graph filters because they are popular architectures for graph deep learning. We theoretically and empirically find that GCNs with a symmetric normalized filter have a within-group preferential attachment (PA) bias in LP. Specifically, GCNs often output LP scores that are proportional to the geometric mean of the (within-group) degrees of the incident nodes when the nodes belong to the same social group.

Our finding can have significant implications for the fairness of GCN LP. For example, consider links within the CS social group in the toy academic collaboration network in Figure 1. Because men in CS, on average, have a higher within-group degree (deg = 3) than women in CS (deg = 1.25), a collaboration recommender system that uses a GCN can suggest men as collaborators at a higher rate. This has the detrimental effect of further concentrating research collaborations among men,

thereby reducing the influence of women in CS and reinforcing their marginalization in the field (Yamamoto & Frachtenberg, 2022).

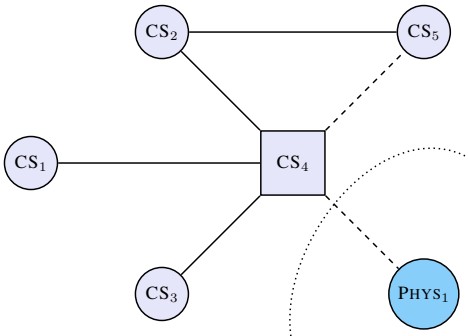

Figure 1: An academic collaboration network where nodes are Computer Science (CS) and Physics (PHYS) researchers, thick edges are current or past collaborations, and dashed edges are collaborations recommended by a GCN. Circular nodes are women and square nodes are men.

Our contributions are as follows:

1. We theoretically uncover that GCNs with a symmetric normalized graph filter have a within-group PA bias in LP (§4.1). We validate our theoretical analysis on diverse real-world network datasets (e.g., citation, collaboration, online social networks) of varying size (§5.1). In doing so, we lay the foundation to study this previously-unexplored PA bias in the GNN setting.
2. We theoretically find that GCNs with a random walk normalized filter may lack a PA bias (§4.3), but empirically show that this is not true (§5.1).
3. We bridge GCN's PA bias with unfairness in LP (§4.2, §5.2). We contribute a new within-group fairness metric for LP, which quantifies disparities in LP scores between social groups, towards combating the amplification of degree and power disparities. To our knowledge, we are the first to study within-group fairness in the GNN setting.
4. We propose a training-time strategy to alleviate within-group unfairness (§4.4), and we assess its effectiveness on citation, online social, and credit networks (§5.3). Our experiments reveal that even for this new form of unfairness, simple regularization approaches can be successful.

## 2 RELATED WORK

**Degree bias in GNNs**   Numerous papers have investigated how GNN performance is degraded for low-degree nodes on node representation learning and classification tasks (Tang et al., 2020; Liu et al., 2021; Kang et al., 2022; Xu et al., 2023; Shomer et al., 2023). Liu et al. (2023) present a generalized notion of degree bias that considers different multi-hop structures around nodes and propose a framework to address it; in contrast to prior work, which focuses on *degree equal opportunity* (i.e., similar accuracy for nodes with the same degree), Liu et al. (2023) also study *degree statistical parity* (i.e., similar prediction rates of each class for nodes with the same degree). Beyond node classification, Wang & Derr (2022) find GNN LP performance disparities across nodes with different degrees: low-degree nodes often benefit from higher performance than high-degree nodes. In this paper, we find that GCNs have a PA bias in LP, and present a new fairness metric which quantifies disparities in GNN LP scores between social groups. We focus on *group fairness* (i.e., parity between social groups) rather than *individual fairness* (i.e., treating similar individuals similarly); this is because producing similar LP scores for similar-degree individuals does not prevent high-degree individuals from unfairly amassing links, and thus power (cf. Figure 1).

**Fair link prediction**   Prior work has investigated the unfairness of GNN LP (Li et al., 2021; Current et al., 2022; Li et al., 2022), often attributing it to graph structure, (e.g., stratification of social groups). However, most of this research has focused on dyadic fairness, i.e., satisfying some notion of parity between inter-group and intra-group links. Like Wang & Derr (2022), we examine how degree bias impacts GNN LP; however, rather than focus on performance disparities across nodes with different degrees, we study GCN's PA bias and LP score disparities across social groups.

**Within-group fairness** Much previous work has studied within-group fairness, i.e., fairness over social subgroups (e.g., Black women, Indigenous men) defined over multiple axes (e.g., race, gender) (Kearns et al., 2017; Foulds et al., 2020; Ghosh et al., 2021; Wang et al., 2022). The motivation of this work is that classifiers can be fair with respect to two social axes separately, but be unfair to subgroups defined over both these axes. While prior research has termed this phenomenon *intersectional* unfairness, we opt for *within-group* unfairness to distinguish it from the critical framework of Intersectionality (Ovalle et al., 2023). We study within-group fairness in the GNN setting.

**Bias and power in networks** A wealth of literature outside fair graph learning has examined how network structure enables discrimination and disparities in capital (Fish et al., 2019; Stoica et al., 2020; Zhang et al., 2021; Bashardoust et al., 2022). Boyd et al. (2014) describe how an individual's position in a social network affects their access to jobs and public health information, as well as how they are surveilled. Stoica et al. (2018) observe that high-degree accounts on Instagram overwhelmingly belong to men and recommendation algorithms further boost these accounts; complementarily, the authors find that even a simple, random walk-based recommendation algorithm can amplify degree disparities between social groups in networks modeled by PA dynamics. Similarly, we investigate how GCN LP can amplify degree disparities in networks and further concentrate power among high-degree individuals.

## 3 PRELIMINARIES

We have a simple, undirected $n$-node graph $\mathcal{G} = (\mathcal{V}, \mathcal{E})$ with self-loops. The nodes have features $(\boldsymbol{x}_i)_{i \in \mathcal{V}}$, with each $\boldsymbol{x}_i \in \mathbb{R}^d$. We denote the adjacency matrix of $\mathcal{G}$ as $\boldsymbol{A} \in \{0, 1\}^{n \times n}$ and the degree matrix as $\boldsymbol{D} = \text{diag}\left(\left(\sum_{j \in \mathcal{V}} \boldsymbol{A}_{ij}\right)_{i \in \mathcal{V}}\right)$, with $\boldsymbol{D} \in \mathbb{N}^{n \times n}$.

We consider two $L$-layer GCN encoders: (1) $\Phi_s : \mathbb{R}^{n \times d} \to \mathbb{R}^{n \times d'}$ (Kipf & Welling, 2016), which uses a symmetric normalized graph filter, and (2) $\Phi_r : \mathbb{R}^{n \times d} \to \mathbb{R}^{n \times d'}$, which uses a random walk normalized filter. $\Phi_s$ and $\Phi_r$ compute node representations as, $\forall i \in \mathcal{V}$:

$$\Phi_s\left((\boldsymbol{x}_j)_{j \in \mathcal{V}}\right)_i = \boldsymbol{s}_i^{(L)}, \qquad\qquad \Phi_r\left((\boldsymbol{x}_j)_{j \in \mathcal{V}}\right)_i = \boldsymbol{r}_i^{(L)} \tag{1}$$

$$\forall l \in [L], \boldsymbol{s}_i^{(l)} = \sigma^{(l)}\left(\sum_{j \in \Gamma(i)} \frac{\boldsymbol{W}_s^{(l)} \boldsymbol{s}_j^{(l-1)}}{\sqrt{\boldsymbol{D}_{ii} \boldsymbol{D}_{jj}}}\right), \qquad \forall l \in [L], \boldsymbol{r}_i^{(l)} = \sigma^{(l)}\left(\sum_{j \in \Gamma(i)} \frac{\boldsymbol{W}_r^{(l)} \boldsymbol{r}_j^{(l-1)}}{\boldsymbol{D}_{ii}}\right), \tag{2}$$

where $\left(\boldsymbol{s}_i^{(0)}\right)_{i \in \mathcal{V}} = \left(\boldsymbol{r}_i^{(0)}\right)_{i \in \mathcal{V}} = (\boldsymbol{x}_i)_{i \in \mathcal{V}}$; $\Gamma(i)$ is the 1-hop neighborhood of $i$; $\boldsymbol{W}_s^{(l)}$ and $\boldsymbol{W}_r^{(l)}$ are the weight matrices corresponding to layer $l$ of $\Phi_s$ and $\Phi_r$, respectively; for $l \in [L-1], \sigma^{(l)}$ is a ReLU non-linearity; and $\sigma^{(L)}$ is the identity function. We now consider the first-order Taylor expansions of $\Phi_s$ and $\Phi_r$ around $(\boldsymbol{0})_{i \in \mathcal{V}}$:

$$\boldsymbol{s}_i^{(L)} = \sum_{j \in \mathcal{V}} \left[\frac{\partial \boldsymbol{s}_i^{(L)}}{\partial \boldsymbol{x}_j}\right] \boldsymbol{x}_j + \xi\left(\boldsymbol{s}_i^{(L)}\right), \quad \boldsymbol{r}_i^{(L)} = \sum_{j \in \mathcal{V}} \left[\frac{\partial \boldsymbol{r}_i^{(L)}}{\partial \boldsymbol{x}_j}\right] \boldsymbol{x}_j + \xi\left(\boldsymbol{r}_i^{(L)}\right), \tag{3}$$

where $\xi$ is the error of the first-order approximations. This error is low when $(\boldsymbol{x}_i)_{i \in \mathcal{V}}$ are close to $\boldsymbol{0}$, which we validate empirically in §5.1. Furthermore, we consider an inner-product LP score function $f_{LP} : \mathbb{R}^{d'} \times \mathbb{R}^{d'} \to \mathbb{R}$ with the form:

$$f_{LP}\left(\boldsymbol{h}_i^{(L)}, \boldsymbol{h}_j^{(L)}\right) = \left(\boldsymbol{h}_i^{(L)}\right)^\mathsf{T} \boldsymbol{h}_j^{(L)} \tag{4}$$

## 4 THEORETICAL ANALYSIS

We leverage spectral graph theory to study how degree bias affects GCN LP. Theoretically, we find that GCNs with a symmetric normalized graph filter have a within-group PA bias (§4.1), but GCNs with a random walk normalized filter may lack such a bias (§4.3). We further bridge GCN's PA bias

with unfairness in GCN LP, proposing a new LP within-group fairness metric (§4.2) and a simple training-time strategy to alleviate unfairness (§4.4). We empirically validate our theoretical results and fairness strategy in §5. We provide proofs for all theoretical results in §A.

Our ultimate goal is to bound the expected LP scores $\mathbb{E}\left[f_{LP}\left(\boldsymbol{s}_i^{(L)}, \boldsymbol{s}_j^{(L)}\right)\right]$ and $\mathbb{E}\left[f_{LP}\left(\boldsymbol{r}_i^{(L)}, \boldsymbol{r}_j^{(L)}\right)\right]$ for nodes $i, j$ in the same social group in terms of the degrees of $i, j$. We begin with Lemma 4.1, which expresses GCN representations (in expectation) as a linear combination of the initial node features. In doing so, we decouple the computation of GCN representations from the non-linearities $\sigma^{(l)}$.

**Lemma 4.1.** *Similarly to Xu et al. (2018), assume that each path from node $i \to j$ in the computation graph of $\Phi_s$ is independently activated with probability $\rho_s(i)$, and similarly, $\rho_r(i)$ for $\Phi_r$. Furthermore, suppose that $\mathbb{E}\left[\xi\left(\boldsymbol{s}_i^{(L)}\right)\right] = \mathbb{E}\left[\xi\left(\boldsymbol{r}_i^{(L)}\right)\right] = \boldsymbol{0}$, where the expectations are taken over the probability distributions of paths activating; our results in §5.1 show that this assumption is reasonable. We define $\alpha_j = \left(\prod_{l=L}^1 \boldsymbol{W}_s^{(l)}\right) \boldsymbol{x}_j$, and $\beta_j = \left(\prod_{l=L}^1 \boldsymbol{W}_r^{(l)}\right) \boldsymbol{x}_j$. Then, $\forall i \in \mathcal{V}$:*

$$\mathbb{E}\left[\boldsymbol{s}_i^{(L)}\right] = \sum_{j \in \mathcal{V}} \rho_s(i) \left(\boldsymbol{D}^{-\frac{1}{2}} \boldsymbol{A} \boldsymbol{D}^{-\frac{1}{2}}\right)_{ij}^L \alpha_j, \quad \mathbb{E}\left[\boldsymbol{r}_i^{(L)}\right] = \sum_{j \in \mathcal{V}} \rho_r(i) \left(\boldsymbol{D}^{-1}\boldsymbol{A}\right)_{ij}^L \beta_j. \quad (5)$$

Lemma 4.1 demonstrates that under certain assumptions, the expected GCN representations can be expressed as a linear combination of the node features that depends on a normalized version of the adjacency matrix (e.g., $\boldsymbol{D}^{-\frac{1}{2}} \boldsymbol{A} \boldsymbol{D}^{-\frac{1}{2}}$, $\boldsymbol{D}^{-1}\boldsymbol{A}$).

We now introduce social groups in $\mathcal{G}$ into our analysis. Suppose that $\mathcal{V}$ can be partitioned into $B$ disjoint social groups $\{S^{(b)}\}_{b \in [B]}$, such that $\bigcup_{b \in [B]} S^{(b)} = \mathcal{V}$ and $\bigcap_{b \in [B]} S^{(b)} = \emptyset$. (If a group comprises $C > 1$ connected components, it can be treated as $C$ separate groups.) Furthermore, we define $\mathcal{G}^{(b)}$ as the induced subgraph of $\mathcal{G}$ formed from $S^{(b)}$. Let $\widehat{\boldsymbol{A}}$ be a within-group adjacency matrix that contains links between nodes in the same group, i.e., $\widehat{\boldsymbol{A}}$ contains the link $(i, j)$ if and only if for some group $S^{(b)}$, $i, j \in S^{(b)}$. Without loss of generality, we reorder the rows and columns of $\widehat{\boldsymbol{A}}$ and $\boldsymbol{A}$ such that $\widehat{\boldsymbol{A}}$ is a block matrix. Let $\widehat{\boldsymbol{D}}$ be the corresponding degree matrix of $\widehat{\boldsymbol{A}}$.

### 4.1 SYMMETRIC NORMALIZED FILTER

We first focus on analyzing $\Phi_s$. We introduce the notation $\boldsymbol{P} = \boldsymbol{D}^{-\frac{1}{2}} \boldsymbol{A} \boldsymbol{D}^{-\frac{1}{2}}$ for the symmetric normalized adjacency matrix. In Lemma 4.2, we present an inequality for the entries of $\boldsymbol{P}^L$ in terms of the spectral properties of $\boldsymbol{A}$. We can then combine this inequality with Lemma 4.1 to bound $\mathbb{E}\left[\boldsymbol{s}_i^{(L)}\right]$, and subsequently $\mathbb{E}\left[f_{LP}\left(\boldsymbol{s}_i^{(L)}, \boldsymbol{s}_j^{(L)}\right)\right]$.

**Lemma 4.2.** *We define $\widehat{\boldsymbol{P}} = \widehat{\boldsymbol{D}}^{-\frac{1}{2}} \widehat{\boldsymbol{A}} \widehat{\boldsymbol{D}}^{-\frac{1}{2}}$, which has the form $\begin{bmatrix} \widehat{\boldsymbol{P}}^{(1)} & & \boldsymbol{0} \\ & \ddots & \\ \boldsymbol{0} & & \widehat{\boldsymbol{P}}^{(B)} \end{bmatrix}$. Each $\widehat{\boldsymbol{P}}^{(b)}$ admits the orthonormal spectral decomposition $\widehat{\boldsymbol{P}}^{(b)} = \sum_{k=1}^{|S^{(b)}|} \lambda_k^{(b)} \boldsymbol{v}_k^{(b)} \left(\boldsymbol{v}_k^{(b)}\right)^\mathsf{T}$. Let $\left(\lambda_k^{(b)}\right)_{1 \le k \le |S^{(b)}|}$ be the eigenvalues of $\widehat{\boldsymbol{P}}^{(b)}$ sorted in non-increasing order; the eigenvalues fall in the range $(-1, 1]$. By the spectral properties of $\widehat{\boldsymbol{P}}^{(b)}$, $\lambda_1^{(b)} = 1$. Following Lovász (2001), we denote the spectral gap of $\widehat{\boldsymbol{P}}^{(b)}$ as $\lambda^{(b)} = \max\left\{\lambda_2^{(b)}, \left|\lambda_{|S^{(b)}|}^{(b)}\right|\right\} < 1$; $\lambda_2^{(b)}$ corresponds to the smallest non-zero eigenvalue of the symmetric normalized graph Laplacian. Let $\boldsymbol{P} = \widehat{\boldsymbol{P}} + \Xi^{(0)}$. If $\mathcal{G}$ is highly modular or approximately disconnected, then $\Xi^{(0)} \approx \boldsymbol{0}$, albeit with negative and non-negative entries. Finally, we define the volume $\mathrm{vol}\left(\mathcal{G}^{(b)}\right) = \sum_{k \in S^{(b)}} \widehat{\boldsymbol{D}}_{kk}$. Then, for $i, j \in S^{(b)}$:*

$$\left|\boldsymbol{P}_{ij}^L - \frac{\sqrt{\widehat{\boldsymbol{D}}_{ii} \widehat{\boldsymbol{D}}_{jj}}}{\mathrm{vol}\left(\mathcal{G}^{(b)}\right)}\right| \le \zeta_s = \left(\lambda^{(b)}\right)^L + \sum_{l=1}^L \binom{L}{l} \left\|\Xi^{(0)}\right\|_{op}^l \left\|\widehat{\boldsymbol{P}}\right\|_{op}^{L-l} \quad (6)$$

*And for* $i \in S^{(b)}, j \notin S^{(b)}$, $\left| \boldsymbol{P}_{ij}^L - 0 \right| \leq \sum_{l=1}^{L} \binom{L}{l} \left\| \Xi^{(0)} \right\|_{op}^l \left\| \widehat{\boldsymbol{P}} \right\|_{op}^{L-l} \leq \zeta_s$.

The proof of Lemma 4.2 is similar to spectral proofs of random walk convergence. When $L$ is small (e.g., 2 for most GCNs (Kipf & Welling, 2016)) and $\left\| \Xi^{(0)} \right\|_{op} \approxeq 0$, $\sum_{l=1}^{L} \binom{L}{l} \left\| \Xi^{(0)} \right\|_{op}^l \left\| \widehat{\boldsymbol{P}} \right\|_{op}^{L-l} \approxeq 0$. Furthermore, with significant stratification between social groups (Hofstra et al., 2017) and high expansion within groups (Malliaros & Megalooikonomou, 2011; Leskovec et al., 2008), $\lambda^{(b)} << 1$. In this case, $\zeta_s \approxeq 0$ and $\boldsymbol{P}_{ij}^L \approxeq \frac{\sqrt{\widehat{\boldsymbol{D}}_{ii}\widehat{\boldsymbol{D}}_{jj}}}{vol(\mathcal{G}^{(b)})}$ for $i, j \in S^{(b)}$. Combining Lemmas 4.1 and 4.2, $\Phi_s$ can oversharpen the expected representations to $\mathbb{E}\left[ \boldsymbol{s}_i^{(L)} \right] \approxeq \rho_s(i) \sqrt{\widehat{\boldsymbol{D}}_{ii}} \cdot \sum_{j \in S^{(b)}} \frac{\sqrt{\widehat{\boldsymbol{D}}_{jj}}}{vol(\mathcal{G}^{(b)})} \alpha_j$ (Keriven, 2022; Giovanni et al., 2022). We use this knowledge to bound $\mathbb{E}\left[ f_{LP}\left( \boldsymbol{s}_i^{(L)}, \boldsymbol{s}_j^{(L)} \right) \right]$ in terms of the degrees of $i, j$.

**Theorem 4.3.** *Following a relaxed assumption from Xu et al. (2018), for nodes* $i, j \in S^{(b)}$, *we assume that* $\rho_s(i) = \rho_s(j) = \overline{\rho}_s(b)$. *Then:*

$$\left| \mathbb{E}\left[ f_{LP}\left( \boldsymbol{s}_i^{(L)}, \boldsymbol{s}_j^{(L)} \right) \right] - \overline{\rho}_s^2(b) \left\| \sum_{k \in S^{(b)}} \frac{\sqrt{\widehat{\boldsymbol{D}}_{kk}}}{vol(\mathcal{G}^{(b)})} \alpha_k \right\|_2^2 \sqrt{\widehat{\boldsymbol{D}}_{ii}\widehat{\boldsymbol{D}}_{jj}} \right| \tag{7}$$

$$\leq \zeta_s \overline{\rho}_s^2(b) \left( \sqrt{\widehat{\boldsymbol{D}}_{ii}} + \sqrt{\widehat{\boldsymbol{D}}_{jj}} \right) \left\| \sum_{k \in S^{(b)}} \frac{\sqrt{\widehat{\boldsymbol{D}}_{kk}}}{vol(\mathcal{G}^{(b)})} \alpha_k \right\|_2 \left( \sum_{k \in \mathcal{V}} \|\alpha_k\|_2 \right) + \zeta_s^2 \overline{\rho}_s^2(b) \left( \sum_{k \in \mathcal{V}} \|\alpha_k\|_2 \right)^2 \tag{8}$$

In simpler terms, Theorem 4.3 states that with social stratification and expansion, the expected LP score $\mathbb{E}\left[ f_{LP}\left( \boldsymbol{s}_i^{(L)}, \boldsymbol{s}_j^{(L)} \right) \right] \propto \sqrt{\widehat{\boldsymbol{D}}_{ii}\widehat{\boldsymbol{D}}_{jj}}$ approximately when $i, j$ belong to the same social group. This is because, as explained before Theorem 4.3, $\zeta_s \approxeq 0$, so the RHS of the bound is $\approxeq 0$. This demonstrates that in LP, GCNs with a symmetric normalized graph filter have a within-group PA bias. If $\Phi_s$ positively influences the formation of links over time, this PA bias can drive "rich get richer" dynamics within social groups (Stoica et al., 2018). As shown in Figure 1 and §4.2, such "rich get richer" dynamics can engender group unfairness when nodes' degrees are statistically associated with their group membership (§4.2). Beyond fairness, Theorem 4.3 reveals that GCNs do not align with theories that *social rank* influences link formation, i.e., the likelihood of a link forming between two nodes is proportional to the *difference* in their degrees (Gu et al., 2018).

## 4.2 WITHIN-GROUP FAIRNESS

We further investigate the fairness implications of the PA bias of $\Phi_s$ in LP. We first introduce an additional set of social groups. Suppose that $\mathcal{V}$ can also be partitioned into $D$ disjoint social groups $\{T^{(d)}\}_{d \in [D]}$; then, we can consider intersections of $\{S^{(b)}\}_{b \in [B]}$ and $\{T^{(d)}\}_{d \in [D]}$. For example, revisiting Figure 1, $S$ may correspond to academic discipline (i.e., CS or PHYS) and $T$ may correspond to gender (i.e., men or women). For simplicity, we let $D = 2$. We measure the unfairness $\Delta^{(b)} : \mathbb{R}^{d'} \times \mathbb{R}^{d'} \to \mathbb{R}$ of LP for group $b$ as:

$$\Delta^{(b)}\left( \boldsymbol{h}_i^{(L)}, \boldsymbol{h}_j^{(L)} \right) := \left| \mathbb{E}_{i,j \sim U((S^{(b)} \cap T^{(1)}) \times S^{(b)})} f_{LP}\left( \boldsymbol{h}_i^{(L)}, \boldsymbol{h}_j^{(L)} \right) - \mathbb{E}_{i,j \sim U((S^{(b)} \cap T^{(2)}) \times S^{(b)})} f_{LP}\left( \boldsymbol{h}_i^{(L)}, \boldsymbol{h}_j^{(L)} \right) \right|, \tag{9}$$

where $U(\cdot)$ is a discrete uniform distribution over the input set. $\Delta^{(b)}$ quantifies disparities in GCN LP scores between $T^{(1)}$ and $T^{(2)}$ within $S^{(b)}$. These LP score disparities can cause degree disparities. Based on Theorem 4.3 and §B.1, when $\zeta_s \approxeq 0$, we can estimate $\Delta^{(b)}\left( \boldsymbol{s}_i^{(L)}, \boldsymbol{s}_j^{(L)} \right)$ as:

$$\widehat{\Delta}^{(b)}\left( \boldsymbol{s}_i^{(L)}, \boldsymbol{s}_j^{(L)} \right) = \frac{\overline{\rho}_s^2(b)}{|S^{(b)}|} \left\| \sum_{k \in S^{(b)}} \frac{\sqrt{\widehat{\boldsymbol{D}}_{kk}}}{vol(\mathcal{G}^{(b)})} \alpha_k \right\|_2^2 \left| \sum_{j \in S^{(b)}} \sqrt{\widehat{\boldsymbol{D}}_{jj}} \left( \underbrace{\mathbb{E}_{i \sim U(S^{(b)} \cap T^{(1)})} \sqrt{\widehat{\boldsymbol{D}}_{ii}} - \mathbb{E}_{i \sim U(S^{(b)} \cap T^{(2)})} \sqrt{\widehat{\boldsymbol{D}}_{ii}}}_{\text{degree disparity}} \right) \right| \tag{10}$$

This suggests that a large disparity in the degree of nodes in $S^{(b)} \cap T^{(1)}$ vs. $S^{(b)} \cap T^{(2)}$ can greatly increase the unfairness $\Delta^{(b)}$ of $\Phi_s$ LP. For example, in Figure 1, the large degree disparity between men and women in CS entails that a GCN collaboration recommender system applied to the network will have a large $\Delta^{(b)}$. We empirically validate these fairness implications on diverse real-world network datasets in §5.2. While we consider pre-activation LP scores in Eqn. 9 (in line with prior work, e.g., Li et al. (2021)), we consider post-sigmoid scores $\sigma\left(f_{LP}\left(\boldsymbol{h}_i^{(L)}, \boldsymbol{h}_j^{(L)}\right)\right)$ (where $\sigma$ is the sigmoid function) in §5.2 and §5.3, as this simulates how LP scores may be processed in practice.

### 4.3 RANDOM WALK NORMALIZED FILTER

We now follow similar steps as with $\Phi_s$ to understand how degree bias affects LP scores for $\Phi_r$.

**Theorem 4.4.** *Let* $\zeta_r = \max_{u,v \in \mathcal{V}} \sqrt{\frac{\widehat{\boldsymbol{D}}_{vv}}{\widehat{\boldsymbol{D}}_{uu}}}\left(\lambda^{(b)}\right)^L + \sum_{l=1}^{L}\binom{L}{l}\left\|\Xi^{(0)}\right\|_{op}^l\left\|\widehat{\boldsymbol{P}}\right\|_{op}^{L-l}$. *Furthermore, for nodes* $i, j \in S^{(b)}$, *assume that* $\rho_r(i) = \rho_r(j) = \overline{\rho}_r(b)$. *Combining Lemmas 4.1 and A.1:*

$$\left| \mathbb{E}\left[f_{LP}\left(\boldsymbol{r}_i^{(L)}, \boldsymbol{r}_j^{(L)}\right)\right] - \overline{\rho}_r^2(b)\left\|\sum_{k \in S^{(b)}}\frac{\widehat{\boldsymbol{D}}_{kk}}{vol(\mathcal{G}^{(b)})}\beta_k\right\|_2^2 \right| \tag{11}$$

$$\leq \zeta_r \overline{\rho}_r^2(b)\left\|\sum_{k \in S^{(b)}}\frac{\widehat{\boldsymbol{D}}_{kk}}{vol(\mathcal{G}^{(b)})}\beta_k\right\|_2\left(\sum_{k \in \mathcal{V}}\|\beta_k\|_2\right) + \zeta_r^2\overline{\rho}_r^2(b)\left(\sum_{k \in \mathcal{V}}\|\beta_k\|_2\right)^2 \tag{12}$$

In other words, if $\zeta_r \cong 0$, $\mathbb{E}\left[f_{LP}\left(\boldsymbol{r}_i^{(L)}, \boldsymbol{r}_j^{(L)}\right)\right]$ is approximately constant when $i, j$ belong to the same social group. Based on Theorem 4.4 and §B.2, we can estimate $\Delta^{(b)}\left(\boldsymbol{s}_i^{(L)}, \boldsymbol{s}_j^{(L)}\right)$ as $\widehat{\Delta}^{(b)}\left(\boldsymbol{s}_i^{(L)}, \boldsymbol{s}_j^{(L)}\right) = 0$. Theoretically, this would suggest that a large disparity in the degree of nodes in $S^{(b)} \cap T^{(1)}$ vs. $S^{(b)} \cap T^{(2)}$ does not increase the unfairness $\Delta^{(b)}$ of $\Phi_r$ LP. However, we find empirically that this is not the case (§5.1).

### 4.4 FAIRNESS REGULARIZER

We propose a simple training-time solution to alleviate within-group LP unfairness regardless of graph filter type and GNN architecture. In particular, we can add a fairness regularization term $\mathcal{L}_{\text{fair}}$ to our original GNN training loss (Kamishima et al., 2011):

$$\mathcal{L}_{\text{new}} = \mathcal{L}_{\text{orig}} + \mathcal{L}_{\text{fair}} = \mathcal{L}_{\text{orig}} + \frac{\lambda_{\text{fair}}}{B}\sum_{b \in [B]}\Delta^{(b)},$$

where $\lambda_{\text{fair}}$ is a tunable hyperparameter that for higher values, pushes the GNN to learn fairer parameters. With our fairness strategy, we empirically observe a significant decrease in $\mathcal{L}_{\text{fair}}$ without a severe drop in LP performance for GCN (§5.3).

## 5 EXPERIMENTS

In this section, we empirically validate our theoretical analysis (§5.1) and the within-group fairness implications of GCN's LP PA bias (§5.2) on diverse real-world network datasets (including citation, credit, collaboration, and online social networks) of varying size. We further find that our simple training-time strategy to alleviate unfairness is effective on citation, online social, and credit networks (§5.3). We include our code and data in the supplementary material. We present experimental results with 4-layer GCN encoders in §G, with similar conclusions.

### 5.1 VALIDATING THEORETICAL ANALYSIS

We validate our theoretical analysis on 10 real-world network datasets (e.g., citation networks, collaboration networks, online social networks), which we describe in §C. Each dataset is natively

intended for node classification; however, we adapt the datasets for LP, treating the connected components within the node classes as the social groups $S^{(b)}$. This design choice is reasonable, as in all the datasets, the classes naturally correspond to socially-relevant groupings of the nodes, or proxies thereof (e.g., in the LastFMAsia dataset, the classes are the home countries of users).

We train GCN encoders $\Phi_s$ and $\Phi_r$ for LP over 10 random seeds (cf. §E for more details). In Figure 2, we plot the theoretic[1] LP score that we derive in §4 against the GCN LP score *for pairs of test nodes belonging to the same social group* (including positive and negative links). In particular, for $\Phi_s$, the theoretic LP score is $\overline{\rho}_s^2(b)\sqrt{\widehat{\boldsymbol{D}}_{ii}\widehat{\boldsymbol{D}}_{jj}}\left\|\sum_{k\in S^{(b)}}\frac{\sqrt{\widehat{\boldsymbol{D}}_{kk}}}{\text{vol}(\mathcal{G}^{(b)})}\alpha_k\right\|_2^2$ and the GCN LP score is $f_{LP}\left(\boldsymbol{s}_i^{(L)},\boldsymbol{s}_j^{(L)}\right)$ (cf. Theorem 4.3). In contrast, for $\Phi_r$, the theoretic LP score is $\overline{\rho}_r^2(b)\left\|\sum_{k\in S^{(b)}}\frac{\widehat{\boldsymbol{D}}_{kk}}{\text{vol}(\mathcal{G}^{(b)})}\beta_k\right\|_2^2$ and the GCN LP score is $f_{LP}\left(\boldsymbol{r}_i^{(L)},\boldsymbol{r}_j^{(L)}\right)$ (cf. Theorem 4.4). For all the datasets, we estimate $\overline{\rho}_s^2(b)$ and $\overline{\rho}_r^2(b)$ separately for each social group $S^{(b)}$ as the slope of the least-squares regression line (through the data from $S^{(b)}$) that predicts the GCN score as a function of the theoretic score. Hence, we do not plot any pair of test nodes that is the only pair in $S^{(b)}$, as it is not possible to estimate $\overline{\rho}_s^2(b)$. The test AUC is further consistently high, indicating that the GCNs are well-trained. The large range of each color in the plots indicates a diversity of LP scores within each social group.

We visually observe that the theoretic LP scores are strong predictors of the $\Phi_s$ scores for each dataset, validating our theoretical analysis. This strength is further confirmed by the generally low NRMSE and high PCC (except for the EN dataset). However, we observe a few cases in which our theoretical analysis does not line up with our experimental results:

1. Our theoretical analysis predicts that the LP score between two nodes $i, j$ that belong to the same social group $S^{(b)}$ will always be non-negative; however, $\Phi_s$ can predict negative scores for pairs of nodes in the same social group. In this case, it appears that $\Phi_s$ relies more on the dissimilarity of (transformed) node features than node degree.
2. For many network datasets (especially from the citation and online social domains), there exist node pairs (near the origin) for which the theoretic LP score underestimates the $\Phi_s$ score. Upon further analysis (cf. Appendix H), we find that the theoretic score is less predictive of the $\Phi_s$ score for nodes $i, j$ when the product of their degrees (i.e., their PA score) or similarity of their features is relatively low.
3. It appears that the theoretic LP score tends to poorly estimate the $\Phi_s$ score when the $\Phi_s$ score is relatively high; this suggests that $\Phi_s$ conservatively relies more on the (dis)similarity of node features than node degree when the degree is large.

We do not observe that the theoretic LP scores are strong predictors of the $\Phi_r$ scores. This could be because the error bound for the theoretic scores for $\Phi_r$, unlike for $\Phi_s$, has an extra dependence $\max_{u,v\in\mathcal{V}}\sqrt{\frac{\widehat{\boldsymbol{D}}_{vv}}{\widehat{\boldsymbol{D}}_{uu}}}$ on the degrees of the incident nodes (cf. $\zeta_r$ in Theorem 4.4). **We explore this further in §I.**

## 5.2 WITHIN-GROUP FAIRNESS

We now empirically validate the implications of GCN's PA bias for within-group unfairness in LP. We run experiments on 3 real-world network datasets: (1) the NBA social network (Dai & Wang, 2021), (2) the German credit network (Agarwal et al., 2021), and (3) a new DBLP-Fairness citation network that we construct. We describe these datasets in §D, including $\{S^{(b)}\}_{b\in[B]}$ and $\{T^{(d)}\}_{d\in[D]}$.

We train 2-layer GCN encoders $\Phi_s$ for LP (cf. §E). In Figure 3, for all the datasets, we plot $\widehat{\Delta}^{(b)}$ vs. $\Delta^{(b)}$ (cf. Eqns. 9, 10) for each $b\in[B]$. We qualitatively and quantitatively observe that $\widehat{\Delta}^{(b)}$ is moderately predictive of $\Delta^{(b)}$ for each dataset. This confirms our theoretical intuition (§4.2) that a large disparity in the degree of nodes in $S^{(b)}\cap T^{(1)}$ vs. $S^{(b)}\cap T^{(2)}$ can greatly increase the unfairness $\Delta^{(b)}$ of $\Phi_s$ LP; such unfairness can amplify degree disparities, worsening power imbalances in the

---

[1]We refer to the score as theoretic because it resulted from our theoretical analysis in §4; we reiterate that our results in §4 rely on the assumptions that we state and the theoretic score is not a ground-truth value.

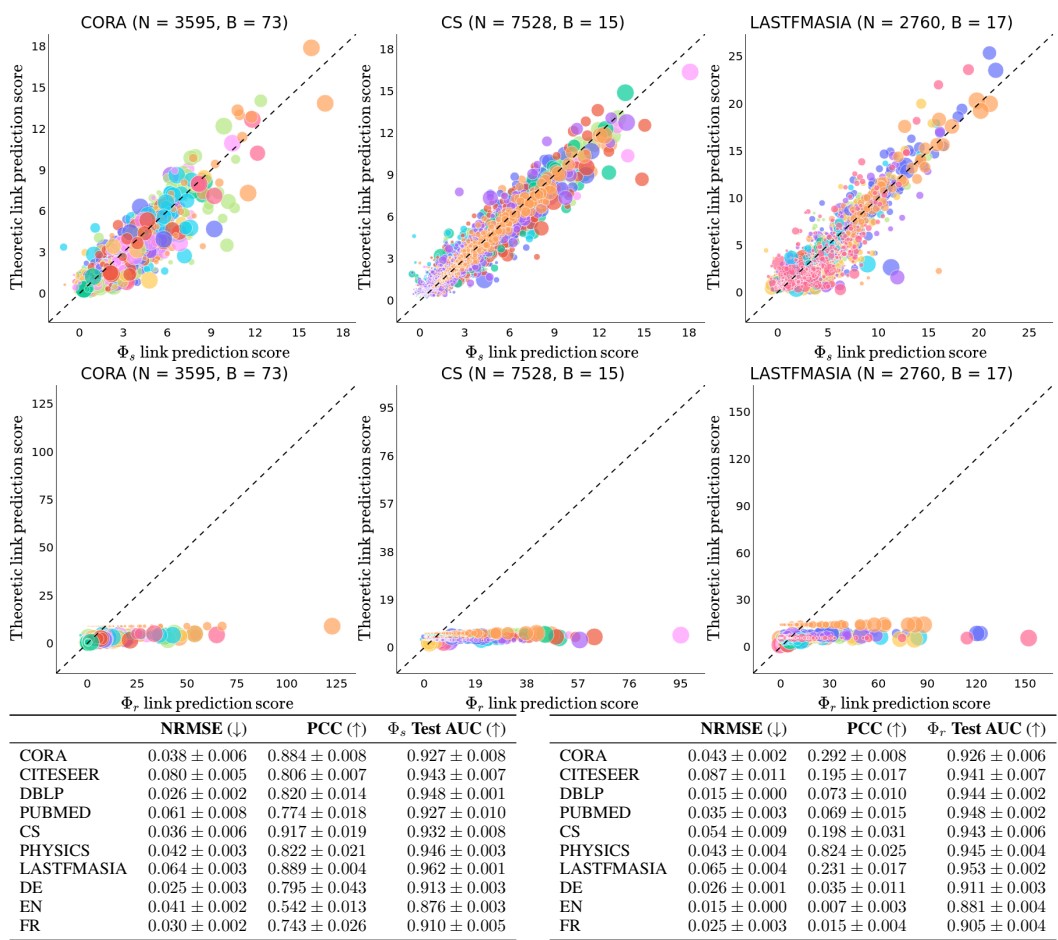

| | **NRMSE** ($\downarrow$) | **PCC** ($\uparrow$) | $\Phi_s$ **Test AUC** ($\uparrow$) | | **NRMSE** ($\downarrow$) | **PCC** ($\uparrow$) | $\Phi_r$ **Test AUC** ($\uparrow$) |
|---|---|---|---|---|---|---|---|
| CORA | $0.038 \pm 0.006$ | $0.884 \pm 0.008$ | $0.927 \pm 0.008$ | CORA | $0.043 \pm 0.002$ | $0.292 \pm 0.008$ | $0.926 \pm 0.006$ |
| CITESEER | $0.080 \pm 0.005$ | $0.806 \pm 0.007$ | $0.943 \pm 0.007$ | CITESEER | $0.087 \pm 0.011$ | $0.195 \pm 0.017$ | $0.941 \pm 0.007$ |
| DBLP | $0.026 \pm 0.002$ | $0.820 \pm 0.014$ | $0.948 \pm 0.001$ | DBLP | $0.015 \pm 0.000$ | $0.073 \pm 0.010$ | $0.944 \pm 0.002$ |
| PUBMED | $0.061 \pm 0.008$ | $0.774 \pm 0.018$ | $0.927 \pm 0.010$ | PUBMED | $0.035 \pm 0.003$ | $0.069 \pm 0.015$ | $0.948 \pm 0.002$ |
| CS | $0.036 \pm 0.006$ | $0.917 \pm 0.019$ | $0.932 \pm 0.008$ | CS | $0.054 \pm 0.009$ | $0.198 \pm 0.031$ | $0.943 \pm 0.006$ |
| PHYSICS | $0.042 \pm 0.003$ | $0.822 \pm 0.021$ | $0.946 \pm 0.003$ | PHYSICS | $0.043 \pm 0.004$ | $0.824 \pm 0.025$ | $0.945 \pm 0.004$ |
| LASTFMASIA | $0.064 \pm 0.003$ | $0.889 \pm 0.004$ | $0.962 \pm 0.001$ | LASTFMASIA | $0.065 \pm 0.004$ | $0.231 \pm 0.017$ | $0.953 \pm 0.002$ |
| DE | $0.025 \pm 0.003$ | $0.795 \pm 0.043$ | $0.913 \pm 0.003$ | DE | $0.026 \pm 0.001$ | $0.035 \pm 0.011$ | $0.911 \pm 0.003$ |
| EN | $0.041 \pm 0.002$ | $0.542 \pm 0.013$ | $0.876 \pm 0.003$ | EN | $0.015 \pm 0.000$ | $0.007 \pm 0.003$ | $0.881 \pm 0.004$ |
| FR | $0.030 \pm 0.002$ | $0.743 \pm 0.026$ | $0.910 \pm 0.005$ | FR | $0.025 \pm 0.003$ | $0.015 \pm 0.004$ | $0.905 \pm 0.004$ |

Figure 2: The plots display the theoretic vs. GCN LP scores for the Cora, CS, and LastFMAsia datasets over 10 random seeds. (We include the plots for the remaining datasets in §F.) The **top row** of plots corresponds to $\Phi_s$, and the **bottom row** to $\Phi_r$. In the plots, each circle corresponds to a single pair of test nodes (between which we are predicting if a link exists). The center of each circle represents the mean of the theoretic and GCN scores and its area captures the range of scores. The color of each circle indicates the social group to which the node pair belongs. The plots include: (1) the total number of test node pairs $N$; (2) the number of social groups $B$; and (3) the line of equality (represented by dashes) for easy comparison of the theoretic and GCN scores. For all the datasets, the tables display: (1) the mean and standard deviation of the GCN test AUC on LP; and (2) the mean and standard deviation of the range-normalized root-mean-square deviation (NRMSE) and Pearson correlation coefficient (PCC) of the theoretic LP scores as predictors of the GCN scores. The **left** table corresponds to $\Phi_s$, and the **right** to $\Phi_r$.

network. Many points deviate from the line of equality; these deviations can be explained by the reasons in §5.1 and the compounding of errors.

## 5.3 FAIRNESS REGULARIZER

We evaluate our solution to alleviate LP unfairness (§4.2). In particular, we add our fairness regularization term $\mathcal{L}_{\text{fair}}$ to the original training loss for the 2-layer $\Phi_s$ and $\Phi_r$ encoders. During each training epoch, we compute $\Delta^{(b)}$ post-sigmoid using only the LP scores over the sampled (positive and negative) training edges. In Table 1, we summarize $\mathcal{L}_{\text{fair}}$ and test AUC for the NBA, German, and DBLP-Fairness datasets with various settings of $\lambda_{\text{fair}}$. For both graph filter types, we generally observe a significant decrease in $\mathcal{L}_{\text{fair}}$ (without a severe drop in test AUC) for $\lambda_{\text{fair}} > 0.0$ over $\lambda_{\text{fair}} = 0.0$; however, the varying magnitudes by which $\mathcal{L}_{\text{fair}}$ decreases across the datasets suggests

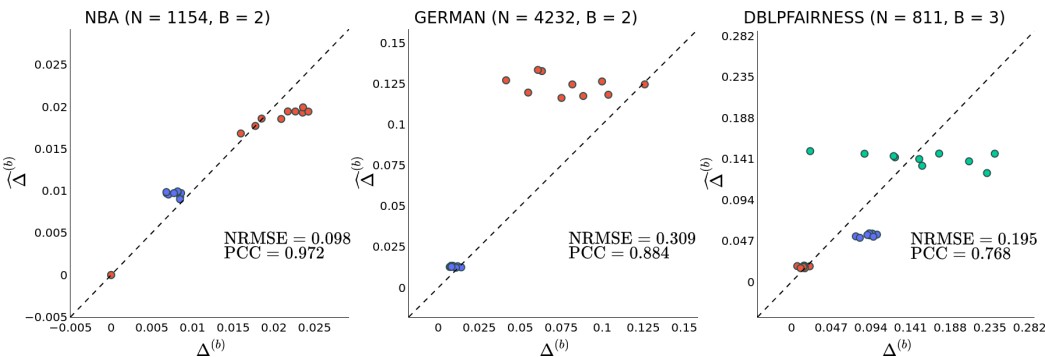

Figure 3: The plots display $\widehat{\Delta}^{(b)}$ vs. $\Delta^{(b)}$ for $\Phi_s$ for the NBA, German, and DBLP-Fairness datasets over all $b \in [B]$ and 10 random seeds. Each point corresponds to a social group $S^{(b)}$, for a different random seed, and the color of the point corresponds to $S^{(b)}$. We compute $\widehat{\Delta}^{(b)}$ and $\Delta^{(b)}$ post-sigmoid using only the LP scores over the sampled (positive and negative) test edges. The plots display the NRMSE and PCC of $\widehat{\Delta}^{(b)}$ as a predictor of $\Delta^{(b)}$.

|  | $\lambda_{\text{fair}}$ | $\mathcal{L}_{\text{fair}}$ ($\downarrow$) | $\Phi_s$ **Test AUC** ($\uparrow$) |  | $\lambda_{\text{fair}}$ | $\mathcal{L}_{\text{fair}}$ ($\downarrow$) | $\Phi_r$ **Test AUC** ($\uparrow$) |
|---|---|---|---|---|---|---|---|
| NBA | 4.0 | $0.000 \pm 0.001$ | $0.753 \pm 0.002$ | NBA | 4.0 | $0.000 \pm 0.001$ | $0.752 \pm 0.002$ |
| NBA | 2.0 | $0.004 \pm 0.003$ | $0.752 \pm 0.003$ | NBA | 2.0 | $0.009 \pm 0.007$ | $0.752 \pm 0.002$ |
| NBA | 1.0 | $0.007 \pm 0.004$ | $0.752 \pm 0.003$ | NBA | 1.0 | $0.016 \pm 0.009$ | $0.752 \pm 0.002$ |
| NBA | 0.0 | $0.013 \pm 0.005$ | $0.752 \pm 0.003$ | NBA | 0.0 | $0.023 \pm 0.012$ | $0.752 \pm 0.002$ |
| DBLPFAIRNESS | 4.0 | $0.072 \pm 0.018$ | $0.741 \pm 0.008$ | DBLPFAIRNESS | 4.0 | $0.100 \pm 0.020$ | $0.762 \pm 0.006$ |
| DBLPFAIRNESS | 2.0 | $0.095 \pm 0.025$ | $0.756 \pm 0.007$ | DBLPFAIRNESS | 2.0 | $0.133 \pm 0.048$ | $0.773 \pm 0.004$ |
| DBLPFAIRNESS | 1.0 | $0.110 \pm 0.033$ | $0.770 \pm 0.010$ | DBLPFAIRNESS | 1.0 | $0.186 \pm 0.054$ | $0.777 \pm 0.005$ |
| DBLPFAIRNESS | 0.0 | $0.145 \pm 0.020$ | $0.778 \pm 0.007$ | DBLPFAIRNESS | 0.0 | $0.221 \pm 0.025$ | $0.776 \pm 0.007$ |
| GERMAN | 4.0 | $0.012 \pm 0.006$ | $0.876 \pm 0.017$ | GERMAN | 4.0 | $0.037 \pm 0.017$ | $0.871 \pm 0.009$ |
| GERMAN | 2.0 | $0.028 \pm 0.017$ | $0.889 \pm 0.017$ | GERMAN | 2.0 | $0.059 \pm 0.016$ | $0.902 \pm 0.005$ |
| GERMAN | 1.0 | $0.038 \pm 0.016$ | $0.897 \pm 0.014$ | GERMAN | 1.0 | $0.071 \pm 0.010$ | $0.907 \pm 0.003$ |
| GERMAN | 0.0 | $0.045 \pm 0.013$ | $0.912 \pm 0.009$ | GERMAN | 0.0 | $0.087 \pm 0.016$ | $0.913 \pm 0.008$ |

Table 1: $\mathcal{L}_{\text{fair}}$ and the test AUC for the NBA, German, and DBLP-Fairness datasets with various settings of $\lambda_{\text{fair}}$. The **left** table corresponds to $\Phi_s$, and the **right** to $\Phi_r$.

that $\lambda_{\text{fair}}$ may need to be tuned per dataset. As expected, we observe a tradeoff between $\mathcal{L}_{\text{fair}}$ and the test AUC as $\lambda_{\text{fair}}$ increases. Our experiments reveal that, regardless of graph filter type, even simple regularization approaches can alleviate this new form of unfairness. As this form of unfairness has not been previously explored, we do not have any baselines.

Our fairness regularizer can be easily integrated into model training, does not require significant additional computation, and directly optimizes for LP fairness. However, it is not applicable in settings where model parameters cannot be retrained or finetuned. Hence, we encourage future research to also explore post-processing fairness strategies. For example, for $\Phi_s$ models, based on our theory (cf. Theorem 4.3), for each pair of nodes $i, j$, we can decay the influence of GCN's PA bias by scaling (pre-activation) LP scores by $\left( \sqrt{\widehat{D}_{ii} \widehat{D}_{jj}} \right)^{-\alpha}$, where $0 < \alpha < 1$ is a hyperparameter that can be tuned to achieve a desirable balance between $\mathcal{L}_{\text{fair}}$ and the test AUC.

## 6 CONCLUSION

We theoretically and empirically show that GCNs can have a PA bias in LP. We analyze how this bias can engender within-group unfairness, and amplify degree and power imbalances in networks. We further propose a simple training-time strategy to alleviate this unfairness. We encourage future work to: (1) explore PA bias in other GNN architectures and directed and heterophilic networks; (2) characterize the "rich get richer" evolution of networks affected by GCN's PA bias; and (3) propose pre-processing and post-processing strategies for within-group LP unfairness.

## ETHICS STATEMENT

We provide our code and data in the supplementary material, along with an MIT license. We include the raw DBLP-Fairness dataset that we construct in the supplementary material, and we detail all data processing steps in §D.3. Our paper touches upon issues of discrimination, bias, and fairness. Throughout, we tie our analysis back to issues of disparity and power, towards advancing justice in graph learning. Some datasets that we use contain protected attribute information (detailed in §D). We avoid using datasets that enable carceral technology (e.g., Recidivism (Agarwal et al., 2021)). While we propose a strategy to alleviate within-group LP unfairness, we emphasize that it is not a 'silver bullet' solution; we encourage graph learning practitioners to adopt a sociotechnical approach and continually adapt their algorithms, datasets, and metrics in response to the everchanging landscape of inequality and power. Furthermore, the fairness of GCN LP should not sidestep concerns about GCN LP being used at all in certain scenarios. We do our best to discuss limitations throughout the paper.

## REPRODUCIBILITY STATEMENT

We provide all our code and data in the supplementary material, along with a README. For each lemma and theorem (§4), all our assumptions are clearly explained and justified either before or in the statement thereof. We include complete proofs of our theoretical claims in §A and §B. We include the raw DBLP-Fairness dataset in the supplementary material, and we detail all data processing steps in §D.3. All our experiments (§5) are run with 10 random seeds and errors are reported. We provide model implementation details in §E.

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

# Supplementary Text

## A    PROOFS

### A.1    PROOF OF LEMMA 4.1

*Proof.* Similarly to Xu et al. (2018); Tang et al. (2020), we compute the first-order partial derivatives of $\Phi_s$ and $\Phi_r$:

$$\frac{\partial \boldsymbol{s}_i^{(L)}}{\partial \boldsymbol{x}_j} = \sum_{p \in \Psi_{i \to j}^{L+1}} \prod_{l=L}^{1} \frac{\text{diag}\left(\mathbb{1}_{\boldsymbol{z}_{p^{(l)}}^{(l)}>0}\right) \boldsymbol{W}_s^{(l)}}{\sqrt{\boldsymbol{D}_{p^{(l)}p^{(l)}} \boldsymbol{D}_{p^{(l-1)}p^{(l-1)}}}}, \quad \frac{\partial \boldsymbol{r}_i^{(L)}}{\partial \boldsymbol{x}_j} = \sum_{p \in \Psi_{i \to j}^{L+1}} \prod_{l=L}^{1} \frac{\text{diag}\left(\mathbb{1}_{\boldsymbol{z}_{p^{(l)}}^{(l)}>0}\right) \boldsymbol{W}_s^{(l)}}{\boldsymbol{D}_{p^{(l)}p^{(l)}}} \tag{13}$$

$$\frac{\partial \boldsymbol{s}_i^{(L)}}{\partial \boldsymbol{x}_j} = \sqrt{\frac{\boldsymbol{D}_{ii}}{\boldsymbol{D}_{jj}}} \sum_{p \in \Psi_{i \to j}^{L+1}} \prod_{l=L}^{1} \frac{\text{diag}\left(\mathbb{1}_{\boldsymbol{z}_{p^{(l)}}^{(l)}>0}\right) \boldsymbol{W}_s^{(l)}}{\boldsymbol{D}_{p^{(l)}p^{(l)}}} \tag{14}$$

where $p^{(l)}$ is the $l$-th node on path $p$ in the computation graph of $\Phi_s$ or $\Phi_r$ ($p^{(L)}$ is node $i$ and $p^{(0)}$ is node $j$); $\Psi_{i \to j}^{\gamma}$ is the set of all $\gamma$-length random walk paths from node $i$ to $j$; and $\boldsymbol{z}_{p^{(l)}}^{(l)}$ is pre-activated $\boldsymbol{s}_{p^{(l)}}^{(l)}$ or $\boldsymbol{r}_{p^{(l)}}^{(l)}$.

With our assumption that the path from node $i \to j$ in the computation graph of $\Phi_s$ is independently activated with probability $\rho_s(i)$, and similarly, $\rho_r(i)$ for $\Phi_r$:

$$\mathbb{E}\left[\frac{\partial \boldsymbol{s}_i^{(L)}}{\partial \boldsymbol{x}_j}\right] = \left(\boldsymbol{D}^{-\frac{1}{2}} \boldsymbol{A} \boldsymbol{D}^{-\frac{1}{2}}\right)_{ij}^{L} \rho_s(i) \left(\prod_{l=L}^{1} \boldsymbol{W}_s^{(l)}\right), \tag{15}$$

$$\mathbb{E}\left[\frac{\partial \boldsymbol{r}_i^{(L)}}{\partial \boldsymbol{x}_j}\right] = \left(\boldsymbol{D}^{-1} \boldsymbol{A}\right)_{ij}^{L} \rho_r(i) \left(\prod_{l=L}^{1} \boldsymbol{W}_r^{(l)}\right). \tag{16}$$

Then, recalling Eqn. 3:

$$\mathbb{E}\left[\boldsymbol{s}_i^{(L)}\right] = \sum_{j \in \mathcal{V}} \left(\boldsymbol{D}^{-\frac{1}{2}} \boldsymbol{A} \boldsymbol{D}^{-\frac{1}{2}}\right)_{ij}^{L} \rho_s(i) \left(\prod_{l=L}^{1} \boldsymbol{W}_s^{(l)}\right) \boldsymbol{x}_j + \boldsymbol{0}, \tag{17}$$

$$\mathbb{E}\left[\boldsymbol{r}_i^{(L)}\right] = \sum_{j \in \mathcal{V}} \left(\boldsymbol{D}^{-1} \boldsymbol{A}\right)_{ij}^{L} \rho_r(i) \left(\prod_{l=L}^{1} \boldsymbol{W}_r^{(l)}\right) \boldsymbol{x}_j + \boldsymbol{0} \tag{18}$$

$$\mathbb{E}\left[\boldsymbol{s}_i^{(L)}\right] = \sum_{j \in \mathcal{V}} \rho_s(i) \left(\boldsymbol{D}^{-\frac{1}{2}} \boldsymbol{A} \boldsymbol{D}^{-\frac{1}{2}}\right)_{ij}^{L} \alpha_j, \quad \mathbb{E}\left[\boldsymbol{r}_i^{(L)}\right] = \sum_{j \in \mathcal{V}} \rho_r(i) \left(\boldsymbol{D}^{-1} \boldsymbol{A}\right)_{ij}^{L} \beta_j. \tag{19}$$

$\square$

### A.2 PROOF OF LEMMA 4.2

*Proof.* For $j \in S^{(b)}$, we can re-express $\widehat{\boldsymbol{P}}_{ij}^L = \left(\widehat{\boldsymbol{P}}^{(b)}\right)_{ij}^L = \left(\boldsymbol{e}^{(i)}\right)^{\mathsf{T}} \left(\widehat{\boldsymbol{P}}^{(b)}\right)^L \boldsymbol{e}^{(j)2}$. By the spectral properties of $\widehat{\boldsymbol{P}}^{(b)}$, $\left(\boldsymbol{e}^{(i)}\right)^{\mathsf{T}} \boldsymbol{v}_1^{(b)} = \sqrt{\frac{\widehat{\boldsymbol{D}}_{ii}}{\text{vol}\left(\mathcal{G}^{(b)}\right)}}$ (Lovász, 2001). Hence:

$$\widehat{\boldsymbol{P}}_{ij}^L = \sum_{k=1}^{\left|S^{(b)}\right|} \left(\lambda_k^{(b)}\right)^L \left(\boldsymbol{e}^{(i)}\right)^{\mathsf{T}} \boldsymbol{v}_k^{(b)} \left(\boldsymbol{v}_k^{(b)}\right)^{\mathsf{T}} \boldsymbol{e}^{(j)} \tag{20}$$

$$= \frac{\sqrt{\widehat{\boldsymbol{D}}_{ii}\widehat{\boldsymbol{D}}_{jj}}}{\text{vol}\left(\mathcal{G}^{(b)}\right)} + \sum_{k=2}^{\left|S^{(b)}\right|} \left(\lambda_k^{(b)}\right)^L \left(\boldsymbol{e}^{(i)}\right)^{\mathsf{T}} \boldsymbol{v}_k^{(b)} \left(\boldsymbol{v}_k^{(b)}\right)^{\mathsf{T}} \boldsymbol{e}^{(j)} \tag{21}$$

Then, by Cauchy-Schwarz:

$$\left|\widehat{\boldsymbol{P}}_{ij}^L - \frac{\sqrt{\widehat{\boldsymbol{D}}_{ii}\widehat{\boldsymbol{D}}_{jj}}}{\text{vol}\left(\mathcal{G}^{(b)}\right)}\right| \leq \left(\lambda^{(b)}\right)^L \sum_{k=1}^{\left|S^{(b)}\right|} \left|\left(\boldsymbol{e}^{(i)}\right)^{\mathsf{T}} \boldsymbol{v}_k^{(b)}\right| \left|\left(\boldsymbol{e}^{(j)}\right)^{\mathsf{T}} \boldsymbol{v}_k^{(b)}\right| \tag{22}$$

$$\leq \left(\lambda^{(b)}\right)^L \left(\sum_{k=1}^{\left|S^{(b)}\right|} \left|\left(\boldsymbol{e}^{(i)}\right)^{\mathsf{T}} \boldsymbol{v}_k^{(b)}\right|^2\right)^{\frac{1}{2}} \left(\sum_{k=1}^{\left|S^{(b)}\right|} \left|\left(\boldsymbol{e}^{(j)}\right)^{\mathsf{T}} \boldsymbol{v}_k^{(b)}\right|^2\right)^{\frac{1}{2}} \tag{23}$$

$$= \left(\lambda^{(b)}\right)^L \left(\left(\boldsymbol{e}^{(i)}\right)^{\mathsf{T}} \boldsymbol{V}^{(b)} \left(\boldsymbol{V}^{(b)}\right)^{\mathsf{T}} \boldsymbol{e}^{(i)}\right)^{\frac{1}{2}} \left(\left(\boldsymbol{e}^{(j)}\right)^{\mathsf{T}} \boldsymbol{V}^{(b)} \left(\boldsymbol{V}^{(b)}\right)^{\mathsf{T}} \boldsymbol{e}^{(j)}\right)^{\frac{1}{2}} \tag{24}$$

$$= \left(\lambda^{(b)}\right)^L \left\|\boldsymbol{e}^{(i)}\right\|_2 \left\|\boldsymbol{e}^{(j)}\right\|_2 \tag{25}$$

$$= \left(\lambda^{(b)}\right)^L \tag{26}$$

Let $\boldsymbol{P}^L = \left(\widehat{\boldsymbol{P}} + \Xi^{(0)}\right)^L = \widehat{\boldsymbol{P}}^L + \Xi^{(L)}$. Then, by the triangle inequality:

$$\left|\boldsymbol{P}_{ij}^L - \frac{\sqrt{\widehat{\boldsymbol{D}}_{ii}\widehat{\boldsymbol{D}}_{jj}}}{\text{vol}\left(\mathcal{G}^{(b)}\right)}\right| \leq \left(\lambda^{(b)}\right)^L + \left|\left(\boldsymbol{e}^{(i)}\right)^{\mathsf{T}} \Xi^{(L)} \boldsymbol{e}^{(j)}\right| \tag{27}$$

$$\leq \left(\lambda^{(b)}\right)^L + \left\|\Xi^{(L)}\right\|_{op} \tag{28}$$

$$\leq \left(\lambda^{(b)}\right)^L + \sum_{l=1}^{L} \binom{L}{l} \left\|\Xi^{(0)}\right\|_{op}^l \left\|\widehat{\boldsymbol{P}}\right\|_{op}^{L-l} \tag{29}$$

For $j \notin S^{(b)}$, $\widehat{\boldsymbol{P}}_{ij}^L = 0$. Then:

$$\left|\boldsymbol{P}_{ij}^L - 0\right| \leq \left|\left(\boldsymbol{e}^{(i)}\right)^{\mathsf{T}} \Xi^{(L)} \boldsymbol{e}^{(j)}\right| \tag{30}$$

$$\leq \sum_{l=1}^{L} \binom{L}{l} \left\|\Xi^{(0)}\right\|_{op}^l \left\|\widehat{\boldsymbol{P}}\right\|_{op}^{L-l} \tag{31}$$

$$\square$$

---

[2]For simplicity, we abuse notation here: $\left(\widehat{\boldsymbol{P}}^{(b)}\right)_{ij}^L$ is not the entry at row $i$ and column $j$, but rather the entry at the row corresponding to node $i$ and column corresponding to node $j$. Similarly, $\boldsymbol{e}^{(i)}$ is the standard basis vector with a 1 at the entry corresponding to node $i$.

## A.3 Proof of Theorem 4.3

*Proof.* For $u, v \in \mathcal{V}$, let $|\delta_{uv}| \leq \zeta_s$. Combining Lemmas 4.1 and 4.2, by our assumption that the computation graph paths to $i, j$ are activated independently:

$$\mathbb{E}\left[f_{LP}\left(\boldsymbol{s}_i^{(L)}, \boldsymbol{s}_j^{(L)}\right)\right] = \mathbb{E}\left[\boldsymbol{s}_i^{(L)}\right]^\mathsf{T}\mathbb{E}\left[\boldsymbol{s}_j^{(L)}\right] \tag{32}$$

$$= \bar{\rho}_s^2(b)\left(\sum_{k \in S^{(b)}}\frac{\sqrt{\widehat{\boldsymbol{D}}_{ii}\widehat{\boldsymbol{D}}_{kk}}}{\text{vol}\left(\mathcal{G}^{(b)}\right)}\alpha_k + \sum_{k \in \mathcal{V}}\delta_{ik}\alpha_k\right)^\mathsf{T}\left(\sum_{k \in S^{(b)}}\frac{\sqrt{\widehat{\boldsymbol{D}}_{jj}\widehat{\boldsymbol{D}}_{kk}}}{\text{vol}\left(\mathcal{G}^{(b)}\right)}\alpha_k + \sum_{k \in \mathcal{V}}\delta_{jk}\alpha_k\right) \tag{33}$$

$$= \bar{\rho}_s^2(b)\sqrt{\widehat{\boldsymbol{D}}_{ii}\widehat{\boldsymbol{D}}_{jj}}\underbrace{\left\|\sum_{k \in S^{(b)}}\frac{\sqrt{\widehat{\boldsymbol{D}}_{kk}}}{\text{vol}(\mathcal{G}^{(b)})}\alpha_k\right\|_2^2}_{\geq 0} \tag{34}$$

$$+ \bar{\rho}_s^2(b)\left(\sqrt{\widehat{\boldsymbol{D}}_{ii}}\sum_{k \in S^{(b)}}\frac{\sqrt{\widehat{\boldsymbol{D}}_{kk}}}{\text{vol}(\mathcal{G}^{(b)})}\alpha_k\right)^\mathsf{T}\left(\sum_{k \in \mathcal{V}}\delta_{jk}\alpha_k\right) \tag{35}$$

$$+ \bar{\rho}_s^2(b)\left(\sum_{k \in \mathcal{V}}\delta_{ik}\alpha_k\right)^\mathsf{T}\left(\sqrt{\widehat{\boldsymbol{D}}_{jj}}\sum_{k \in S^{(b)}}\frac{\sqrt{\widehat{\boldsymbol{D}}_{kk}}}{\text{vol}(\mathcal{G}^{(b)})}\alpha_k\right) \tag{36}$$

$$+ \bar{\rho}_s^2(b)\left(\sum_{k \in \mathcal{V}}\delta_{ik}\alpha_k\right)^\mathsf{T}\left(\sum_{k \in \mathcal{V}}\delta_{jk}\alpha_k\right) \tag{37}$$

Then, by Cauchy-Schwarz and the triangle inequality:

$$\left|\mathbb{E}\left[f_{LP}\left(\boldsymbol{s}_i^{(L)}, \boldsymbol{s}_j^{(L)}\right)\right] - \underbrace{\bar{\rho}_s^2(b)\left\|\sum_{k \in S^{(b)}}\frac{\sqrt{\widehat{\boldsymbol{D}}_{kk}}}{\text{vol}(\mathcal{G}^{(b)})}\alpha_k\right\|_2^2\sqrt{\widehat{\boldsymbol{D}}_{ii}\widehat{\boldsymbol{D}}_{jj}}}_{\propto\sqrt{\widehat{\boldsymbol{D}}_{ii}\widehat{\boldsymbol{D}}_{jj}}}\right| \tag{38}$$

$$\leq \zeta_s\bar{\rho}_s^2(b)\left(\sqrt{\widehat{\boldsymbol{D}}_{ii}} + \sqrt{\widehat{\boldsymbol{D}}_{jj}}\right)\left\|\sum_{k \in S^{(b)}}\frac{\sqrt{\widehat{\boldsymbol{D}}_{kk}}}{\text{vol}(\mathcal{G}^{(b)})}\alpha_k\right\|_2\left(\sum_{k \in \mathcal{V}}\|\alpha_k\|_2\right) + \zeta_s^2\bar{\rho}_s^2(b)\left(\sum_{k \in \mathcal{V}}\|\alpha_k\|_2\right)^2 \tag{39}$$

$$\square$$

## A.4  Lemma A.1

**Lemma A.1.** *We introduce the notation* $\boldsymbol{P} = \boldsymbol{D}^{-1}\boldsymbol{A}$. *We further define* $\widehat{\boldsymbol{P}} = \widehat{\boldsymbol{D}}^{-1}\widehat{\boldsymbol{A}}$. *Fix* $i \in S^{(b)}$.
*Then, for* $j \in S^{(b)}$:

$$\left| \boldsymbol{P}_{ij}^L - \frac{\widehat{\boldsymbol{D}}_{jj}}{vol\left(\mathcal{G}^{(b)}\right)} \right| \leq \sqrt{\frac{\widehat{\boldsymbol{D}}_{jj}}{\widehat{\boldsymbol{D}}_{ii}}} \left(\lambda^{(b)}\right)^L + \sum_{l=1}^{L} \binom{L}{l} \left\|\Xi^{(0)}\right\|_{op}^l \left\|\widehat{\boldsymbol{P}}\right\|_{op}^{L-l} \tag{40}$$

*And for* $j \notin S^{(b)}$:

$$\left| \boldsymbol{P}_{ij}^L - 0 \right| \leq \sum_{l=1}^{L} \binom{L}{l} \left\|\Xi^{(0)}\right\|_{op}^l \left\|\widehat{\boldsymbol{P}}\right\|_{op}^{L-l} \tag{41}$$

*Proof.* Similar to the proof of Lemma 4.2:

$$\widehat{\boldsymbol{P}}_{ij}^L = \frac{\widehat{\boldsymbol{D}}_{jj}}{\text{vol}\left(\mathcal{G}^{(b)}\right)} + \sqrt{\frac{\widehat{\boldsymbol{D}}_{jj}}{\widehat{\boldsymbol{D}}_{ii}}} \sum_{k=2}^{\left|S^{(b)}\right|} \left(\lambda_k^{(b)}\right)^L \left(\boldsymbol{e}^{(i)}\right)^\mathsf{T} \boldsymbol{v}_k^{(b)} \left(\boldsymbol{v}_k^{(b)}\right)^\mathsf{T} \boldsymbol{e}^{(j)} \tag{42}$$

Subsequently:

$$\left| \widehat{\boldsymbol{P}}_{ij}^L - \frac{\widehat{\boldsymbol{D}}_{jj}}{\text{vol}\left(\mathcal{G}^{(b)}\right)} \right| \leq \sqrt{\frac{\widehat{\boldsymbol{D}}_{jj}}{\widehat{\boldsymbol{D}}_{ii}}} \left(\lambda^{(b)}\right)^L \tag{43}$$

Finally:

$$\left| \boldsymbol{P}_{ij}^L - \frac{\widehat{\boldsymbol{D}}_{jj}}{\text{vol}\left(\mathcal{G}^{(b)}\right)} \right| \leq \zeta_r = \max_{u,v \in \mathcal{V}} \sqrt{\frac{\widehat{\boldsymbol{D}}_{vv}}{\widehat{\boldsymbol{D}}_{uu}}} \left(\lambda^{(b)}\right)^L + \sum_{l=1}^{L} \binom{L}{l} \left\|\Xi^{(0)}\right\|_{op}^l \left\|\widehat{\boldsymbol{P}}\right\|_{op}^{L-l} \tag{44}$$

For $j \notin S^{(b)}$, $\widehat{\boldsymbol{P}}_{ij}^L = 0$. Then:

$$\left| \boldsymbol{P}_{ij}^L - 0 \right| \leq \sum_{l=1}^{L} \binom{L}{l} \left\|\Xi^{(0)}\right\|_{op}^l \left\|\widehat{\boldsymbol{P}}\right\|_{op}^{L-l} \leq \zeta_r \tag{45}$$

$$\square$$

## A.5 PROOF OF THEOREM 4.4

*Proof.* For $u, v \in \mathcal{V}$, let $|\delta_{uv}| \leq \zeta_r$. Combining Lemmas 4.1 and A.1, by our assumption that the computation graph paths to $i, j$ are activated independently:

$$\mathbb{E}\left[f_{LP}\left(\boldsymbol{r}_i^{(L)}, \boldsymbol{r}_j^{(L)}\right)\right] = \mathbb{E}\left[\boldsymbol{r}_i^{(L)}\right]^\mathsf{T} \mathbb{E}\left[\boldsymbol{r}_j^{(L)}\right] \tag{46}$$

$$= \overline{\rho}_r^2(b) \left(\sum_{k \in S^{(b)}} \frac{\widehat{\boldsymbol{D}}_{kk}}{\text{vol}\left(\mathcal{G}^{(b)}\right)} \beta_k + \sum_{k \in \mathcal{V}} \delta_{ik} \beta_k\right)^\mathsf{T} \left(\sum_{k \in S^{(b)}} \frac{\widehat{\boldsymbol{D}}_{kk}}{\text{vol}\left(\mathcal{G}^{(b)}\right)} \beta_k + \sum_{k \in \mathcal{V}} \delta_{jk} \beta_k\right) \tag{47}$$

$$= \overline{\rho}_r^2(b) \underbrace{\left\|\sum_{k \in S^{(b)}} \frac{\widehat{\boldsymbol{D}}_{kk}}{\text{vol}(\mathcal{G}^{(b)})} \beta_k\right\|_2^2}_{\geq 0} + \overline{\rho}_r^2(b) \left(\sum_{k \in S^{(b)}} \frac{\widehat{\boldsymbol{D}}_{kk}}{\text{vol}(\mathcal{G}^{(b)})} \beta_k\right)^\mathsf{T} \left(\sum_{k \in \mathcal{V}} \delta_{jk} \beta_k\right) \tag{48}$$

$$+ \overline{\rho}_r^2(b) \left(\sum_{k \in \mathcal{V}} \delta_{ik} \beta_k\right)^\mathsf{T} \left(\sum_{k \in S^{(b)}} \frac{\widehat{\boldsymbol{D}}_{kk}}{\text{vol}(\mathcal{G}^{(b)})} \beta_k\right) + \overline{\rho}_r^2(b) \left(\sum_{k \in \mathcal{V}} \delta_{ik} \beta_k\right)^\mathsf{T} \left(\sum_{k \in \mathcal{V}} \delta_{jk} \beta_k\right) \tag{49}$$

Then, by Cauchy-Schwarz and the triangle inequality:

$$\left|\mathbb{E}\left[f_{LP}\left(\boldsymbol{r}_i^{(L)}, \boldsymbol{r}_j^{(L)}\right)\right] - \overline{\rho}_r^2(b) \underbrace{\left\|\sum_{k \in S^{(b)}} \frac{\widehat{\boldsymbol{D}}_{kk}}{\text{vol}(\mathcal{G}^{(b)})} \beta_k\right\|_2^2}_{\propto \text{ constant}}\right| \tag{50}$$

$$\leq \zeta_r \overline{\rho}_r^2(b) \left\|\sum_{k \in S^{(b)}} \frac{\widehat{\boldsymbol{D}}_{kk}}{\text{vol}(\mathcal{G}^{(b)})} \beta_k\right\|_2 \left(\sum_{k \in \mathcal{V}} \|\beta_k\|_2\right) + \zeta_r^2 \overline{\rho}_r^2(b) \left(\sum_{k \in \mathcal{V}} \|\beta_k\|_2\right)^2 \tag{51}$$

$$\square$$

# B APPROXIMATION OF $\Delta^{(b)}$

## B.1 APPROXIMATION OF $\Delta^{(b)}$ FOR $\Phi_s$

$$\Delta^{(b)} \left( \boldsymbol{s}_i^{(L)}, \boldsymbol{s}_j^{(L)} \right) \tag{52}$$

$$= \Big| \frac{1}{|(S^{(b)} \cap T^{(1)}) \times S^{(b)}|} \sum_{i \in S^{(b)} \cap T^{(1)}} \sum_{j \in S^{(b)}} f_{LP} \left( \boldsymbol{s}_i^{(L)}, \boldsymbol{s}_j^{(L)} \right) \tag{53}$$

$$- \frac{1}{|(S^{(b)} \cap T^{(2)}) \times S^{(b)}|} \sum_{i \in S^{(b)} \cap T^{(2)}} \sum_{j \in S^{(b)}} f_{LP} \left( \boldsymbol{s}_i^{(L)}, \boldsymbol{s}_j^{(L)} \right) \Big| \tag{54}$$

$$\approx \Big| \frac{1}{|S^{(b)} \cap T^{(1)}| |S^{(b)}|} \sum_{i \in S^{(b)} \cap T^{(1)}} \sum_{j \in S^{(b)}} \overline{\rho}_s^2(b) \sqrt{\widehat{\boldsymbol{D}}_{ii} \widehat{\boldsymbol{D}}_{jj}} \left\| \sum_{k \in S^{(b)}} \frac{\sqrt{\widehat{\boldsymbol{D}}_{kk}}}{\text{vol}(\mathcal{G}^{(b)})} \alpha_k \right\|_2^2 \tag{55}$$

$$- \frac{1}{|S^{(b)} \cap T^{(2)}| |S^{(b)}|} \sum_{i \in S^{(b)} \cap T^{(2)}} \sum_{j \in S^{(b)}} \overline{\rho}_s^2(b) \sqrt{\widehat{\boldsymbol{D}}_{ii} \widehat{\boldsymbol{D}}_{jj}} \left\| \sum_{k \in S^{(b)}} \frac{\sqrt{\widehat{\boldsymbol{D}}_{kk}}}{\text{vol}(\mathcal{G}^{(b)})} \alpha_k \right\|_2^2 \Big| \tag{56}$$

$$= \frac{\overline{\rho}_s^2(b)}{|S^{(b)}|} \left\| \sum_{k \in S^{(b)}} \frac{\sqrt{\widehat{\boldsymbol{D}}_{kk}}}{\text{vol}(\mathcal{G}^{(b)})} \alpha_k \right\|_2^2 \left| \sum_{j \in S^{(b)}} \sqrt{\widehat{\boldsymbol{D}}_{jj}} \underbrace{\left( \underset{i \sim U(S^{(b)} \cap T^{(1)})}{\mathbb{E}} \sqrt{\widehat{\boldsymbol{D}}_{ii}} - \underset{i \sim U(S^{(b)} \cap T^{(2)})}{\mathbb{E}} \sqrt{\widehat{\boldsymbol{D}}_{ii}} \right)}_{\text{degree disparity}} \right| \tag{57}$$

## B.2 APPROXIMATION OF $\Delta^{(b)}$ FOR $\Phi_r$

$$\Delta^{(b)} \left( \boldsymbol{r}_i^{(L)}, \boldsymbol{r}_j^{(L)} \right) \tag{58}$$

$$= \Big| \frac{1}{|(S^{(b)} \cap T^{(1)}) \times S^{(b)}|} \sum_{i \in S^{(b)} \cap T^{(1)}} \sum_{j \in S^{(b)}} f_{LP} \left( \boldsymbol{r}_i^{(L)}, \boldsymbol{r}_j^{(L)} \right) \tag{59}$$

$$- \frac{1}{|(S^{(b)} \cap T^{(2)}) \times S^{(b)}|} \sum_{i \in S^{(b)} \cap T^{(2)}} \sum_{j \in S^{(b)}} f_{LP} \left( \boldsymbol{r}_i^{(L)}, \boldsymbol{r}_j^{(L)} \right) \Big| \tag{60}$$

$$\approx \Big| \frac{1}{|S^{(b)} \cap T^{(1)}| |S^{(b)}|} \sum_{i \in S^{(b)} \cap T^{(1)}} \sum_{j \in S^{(b)}} \overline{\rho}_r^2(b) \left\| \sum_{k \in S^{(b)}} \frac{\widehat{\boldsymbol{D}}_{kk}}{\text{vol}(\mathcal{G}^{(b)})} \beta_k \right\|_2^2 \tag{61}$$

$$- \frac{1}{|S^{(b)} \cap T^{(2)}| |S^{(b)}|} \sum_{i \in S^{(b)} \cap T^{(2)}} \sum_{j \in S^{(b)}} \overline{\rho}_r^2(b) \left\| \sum_{k \in S^{(b)}} \frac{\widehat{\boldsymbol{D}}_{kk}}{\text{vol}(\mathcal{G}^{(b)})} \beta_k \right\|_2^2 \Big| \tag{62}$$

$$= 0 \tag{63}$$

## C  DATASETS USED IN §5.1

In our experiments in §5.1, we use 10 real-world network datasets from Bojchevski & Günnemann (2018), Shchur et al. (2018), Rozemberczki & Sarkar (2020), and Rozemberczki et al. (2021), covering diverse domains (e.g., citation networks, collaboration networks, online social networks). We provide a description and some statistics of each dataset in Table 2. All the datasets have node features and are undirected.

- In all the citation network datasets, nodes represent documents, edges represent citation links, and features are a bag-of-words representation of documents. We row-normalize the features to sum to 1, following Fey & Lenssen (2019)[3]. The classification task is to predict the topic of documents.

- In the collaboration network datasets, nodes represent authors, edges represent coauthorships, and features are embeddings of paper keywords for authors' papers. The classification task is to predict the most active field of study for authors.

- In the LastFMAsia network dataset, nodes represent LastFM users from Asia, edges represent friendships between users, and features are embeddings of the artists liked by users. The classification task is to predict the home country of users.

- In the Twitch network datasets, nodes represent gamers on Twitch, edges represent followerships between them, and features are embeddings of the history of games played by the Twitch users. The classification task is to predict whether or not a gamer streams adult content.

We only run experiments on datasets that can fit without sampling nodes on a single NVIDIA GeForce GTX Titan Xp Graphic Card with 12196MiB of space. Furthermore, we only consider the three largest datasets (i.e., with the most nodes) from Rozemberczki et al. (2021). We use PyTorch Geometric to load and process all datasets (Fey & Lenssen, 2019).

| Name | Domain | # Nodes | # Edges | # Features | # Classes |
|------|--------|---------|---------|-----------|-----------|
| Cora | citation | 19793 | 126842 | 8710 | 70 |
| CiteSeer | citation | 4230 | 10674 | 602 | 6 |
| DBLP | citation | 17716 | 105734 | 1639 | 4 |
| PubMed | citation | 19717 | 88648 | 500 | 3 |
| CS | collaboration | 18333 | 163788 | 6805 | 15 |
| Physics | collaboration | 34493 | 495924 | 8415 | 5 |
| LastFMAsia | online social | 7624 | 55612 | 128 | 18 |
| Twitch-DE | online social | 9498 | 315774 | 128 | 2 |
| Twitch-EN | online social | 7126 | 77774 | 128 | 2 |
| Twitch-FR | online social | 6551 | 231883 | 128 | 2 |

Table 2: Summary of the datasets used in our experiments.

---

[3]https://github.com/pyg-team/pytorch_geometric/blob/master/examples/link_pred.py

## D  DATASETS USED IN §5.2

We run experiments on 3 real-world network datasets: (1) the NBA social network (cf. §D.1), (2) the German credit network (cf. §D.2), and (3) a new DBLP-Fairness citation network that we construct (cf. §D.3). All the datasets have node features and are undirected. We do not pass sensitive attributes as features to the models that we train. For all the datasets, we min-max normalize node features to fall in $[-1, 1]$, following Dai & Wang (2021) and Agarwal et al. (2021). For all datasets, $D = 2$.

### D.1  NBA DATASET

The NBA network (Dai & Wang, 2021) has 403 nodes representing NBA basketball players who are connected if they follow each other on Twitter. There are 21242 links. Each node has 95 features, with an average degree of $52.71 \pm 35.14$. We consider two sensitive attributes per node:

- Age $\{S^{(b)}\}_{b \in [B]}$: how old the payer is, i.e., YOUNG ($\leq 25$ years) or OLD ($> 25$ years).
- Nationality $\{T^{(d)}\}_{d \in [D]}$: from where the player is, i.e., UNITED STATES or OVERSEAS.

### D.2  GERMAN DATASET

The German network (Agarwal et al., 2021) comprises 1000 nodes representing clients in a German bank who are connected if they have similar credit accounts. The German network is not natively a graph dataset; synthetic edges were created by Agarwal et al.. There are 44484 links. Each node has 27 features (e.g., loan amount, account-related features), with an average degree of $44.48 \pm 26.52$. We consider two sensitive attributes per node:

- Foreign worker $\{S^{(b)}\}_{b \in [B]}$: whether the client is a foreign worker, i.e., YES or NO.
- Gender $\{T^{(d)}\}_{d \in [D]}$: the gender with which the client identifies, i.e., MAN or WOMAN.

### D.3  DBLP-FAIRNESS DATASET

In this subsection, we detail how we construct the DBLP-Fairness dataset. We build DBLP-Fairness, as there are only a few natively-graph network datasets with sensitive attributes that are appropriate for graph learning (Subramonian et al., 2022).

We begin with the version of the DBLP-Citation-network V12 dataset from (Tang et al., 2008) that was processed by Xu et al. (2021). This dataset has 3658127 nodes. Each node represents a paper and each edge represents a citation link. We consider five node features:

- Team size: the number of authors on the paper.
- Mean collaborators: the average number of collaborators with whom the authors have previously published.
- Gini collaborators: the Gini coefficient of the number of collaborators with whom the authors have previously published.
- Mean productivity: the average number of papers that the authors have previously published.
- Gini productivity: the Gini coefficient of the number of papers that the authors have previously published.

We also consider two sensitive attributes per node:

- Field $\{S^{(b)}\}_{b \in [B]}$: the field to which the paper belongs, i.e., PROGRAMMING LANGUAGES or DATABASES.
- Nationality $\{T^{(d)}\}_{d \in [D]}$: the country where most authors reside, i.e., UNITED STATES or CHINA.

In DBLP-Fairness, we only include papers whose nationality is UNITED STATES or CHINA; we do this, as American and Chinese citation networks are known to be stratified (Zhao et al., 2022).

We also only include papers whose field is PROGRAMMING LANGUAGES or DATABASES; we infer the field of a paper using its keywords (i.e., whether they contain "programming language" and "database"), and discard papers which include both 'programming language" and "database" in its keywords. Furthermore, we filter out all papers from before 2010. Our filtering choices with regards to field and year may appear arbitrary; however, we sought DBLB-Fairness to be of comparable size to the citation networks in §C. Following filtering, we were left with 14537 nodes and 24844 edges.

# E  MODELS

For all experiments, we use GCN encoders (Kipf & Welling, 2017) to get node representations. Each encoder has two layers (128-dimensional hidden layer, 64-dimensional output layer) with a ReLU nonlinearity in between. We only use two layers, as this is common practice in graph deep learning to prevent oversmoothing (Oono & Suzuki, 2020); however, we run experiments with four layers in §G. We do not use any regularization (e.g., Dropout, BatchNorm). The encoders are explicitly trained for LP with the inner-product LP score function in Eqn. 4, binary cross-entropy loss, and the Adam optimizer with full-batch gradient descent and a learning rate of 0.01 (Kingma & Ba, 2014). We use a random link split of 0.85-0.05-0.1 for train-val-test, following the PyTorch Geometric LP example[4]. We train the encoders for 100 epochs, with a new round of negative link sampling during every epoch; we use a 1:1 ratio of positive to negative links. We ultimately select the model parameters with the highest validation ROC-AUC. Although we do not do any hyperparameter tuning, the test ROC-AUC values (displayed in the figures in §5) indicate that the encoders are well-trained. We use PyTorch (Paszke et al., 2019) and PyTorch Geometric (Fey & Lenssen, 2019) to train all the encoders on a single NVIDIA GeForce GTX Titan Xp Graphic Card with 12196MiB of space.

---

[4]`https://github.com/pyg-team/pytorch_geometric/blob/master/examples/link_pred.py`

# F  REMAINING PLOTS

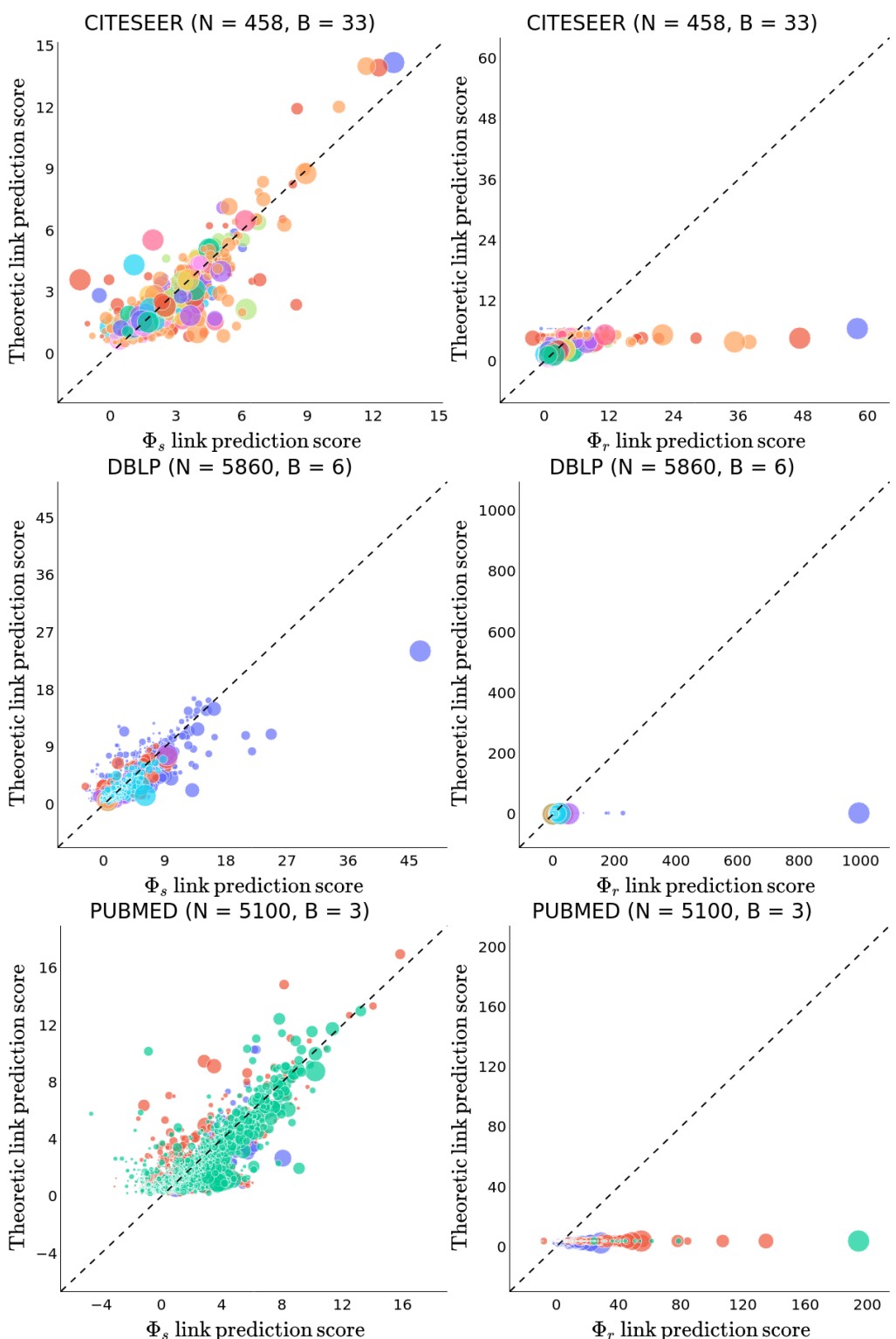

Figure 4: Theoretic vs. GCN LP scores for citation network datasets.

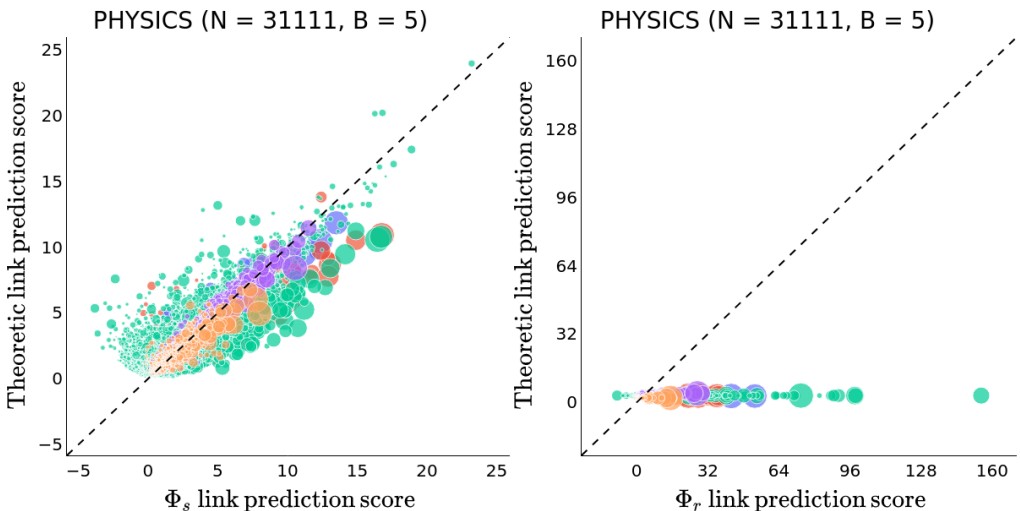

Figure 5: Theoretic vs. GCN LP scores for collaboration network datasets.

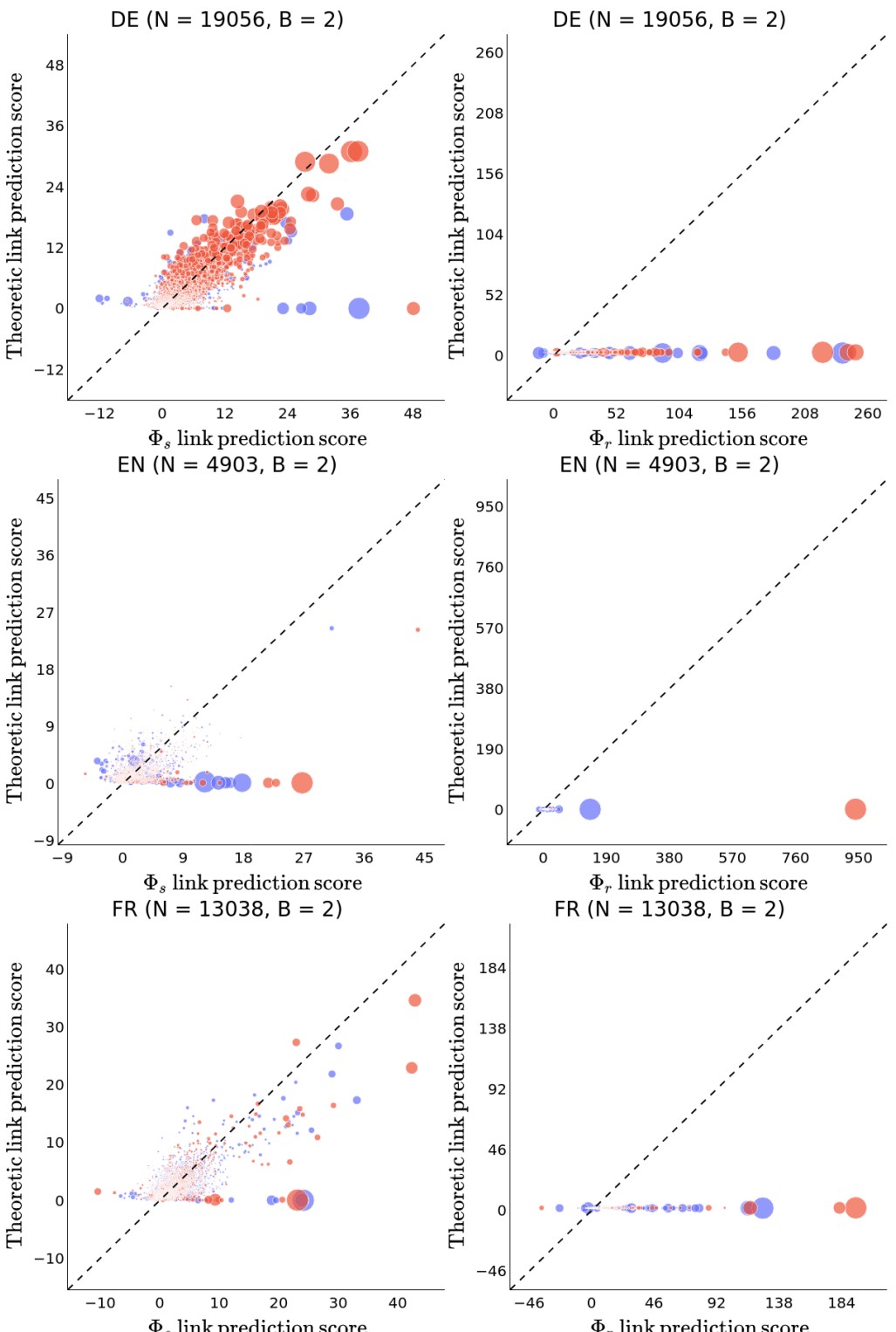

Figure 6: Theoretic vs. GCN LP scores for online social network datasets.

# G  ADDITIONAL EXPERIMENTS

## G.1  ADDITIONAL EXPERIMENTS FOR §5.1 (4-LAYER ENCODERS)

We run the experiments from §5.1 for $\Phi_s$ with the same settings, except we use 4-layer (instead of 2-layer) encoders (128-dimensional hidden layers, 64-dimensional output layer). We run these additional experiments because the error bound for the theoretic LP scores for $\Phi_s$ depends on the number of encoder layers $L$. We find that that the experimental results continue to support our theoretical analysis, both qualitatively and quantitatively (cf. Table 3, Figure 7); the NRMSE and PCC values are comparable to or better than those from the experiments with the 2-layer encoders (especially for the EN dataset).

| | NRMSE ($\downarrow$) | PCC ($\uparrow$) | $\Phi_s$ Test AUC ($\uparrow$) |
|---|---|---|---|
| CORA | $0.044 \pm 0.006$ | $0.858 \pm 0.026$ | $0.853 \pm 0.028$ |
| CITESEER | $0.057 \pm 0.006$ | $0.890 \pm 0.017$ | $0.861 \pm 0.026$ |
| DBLP | $0.021 \pm 0.002$ | $0.885 \pm 0.054$ | $0.887 \pm 0.019$ |
| PUBMED | $0.056 \pm 0.009$ | $0.802 \pm 0.024$ | $0.900 \pm 0.006$ |
| CS | $0.039 \pm 0.006$ | $0.918 \pm 0.008$ | $0.949 \pm 0.004$ |
| PHYSICS | $0.030 \pm 0.002$ | $0.077 \pm 0.013$ | $0.950 \pm 0.004$ |
| LASTFMASIA | $0.040 \pm 0.004$ | $0.938 \pm 0.005$ | $0.949 \pm 0.002$ |
| DE | $0.014 \pm 0.003$ | $0.918 \pm 0.025$ | $0.882 \pm 0.002$ |
| EN | $0.034 \pm 0.005$ | $0.752 \pm 0.036$ | $0.846 \pm 0.008$ |
| FR | $0.019 \pm 0.003$ | $0.833 \pm 0.038$ | $0.896 \pm 0.003$ |

Table 3: The test AUC of the 4-layer $\Phi_s$ encoders on the real-world network datasets, and the NRMSE and PCC of the theoretic LP scores as predictors of the $\Phi_s$ scores.

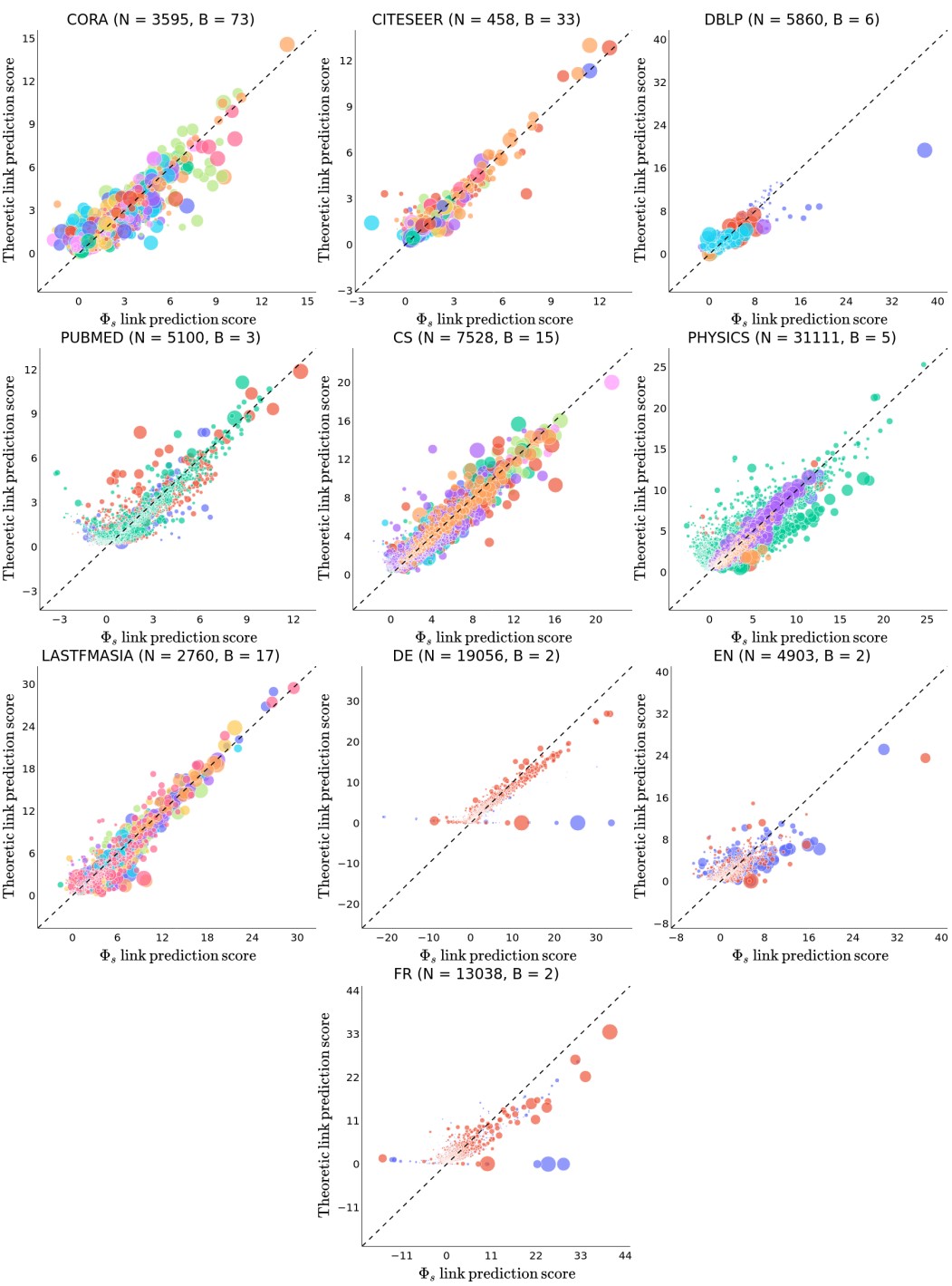

Figure 7: Theoretic LP score vs. 4-layer $\Phi_s$ LP score for all network datasets.

### G.2 Additional experiments for §5.1 (Hadamard Product and MLP LP score function)

We also run the experiments from §5.1 for $\Phi_s$ with the same settings, except we use the following LP score function:

$$f_{LP}\left(\boldsymbol{h}_i^{(L)}, \boldsymbol{h}_j^{(L)}\right) = f_{MLP}\left(\boldsymbol{h}_i^{(L)} \odot \boldsymbol{h}_j^{(L)}\right),$$

where $\odot$ is the Hadamard product and $f_{MLP}$ is a 2-layer MLP with a 64-dimensional hidden layer and ReLU nonlinearity. We run these additional experiments because a Hadamard product and MLP score function is often used in the literature. We find that that our theoretical analysis is still relevant to and reasonably supports the experimental results, both qualitatively and quantitatively (cf. Table 4, Figure 8). This could be because MLPs have an inductive bias towards learning simpler, often linear functions (Nakkiran et al., 2019; Valle-Pérez et al., 2019), and our theoretical findings are generalizable to linear LP score functions. Notably, in this setting, $\Phi_s$ makes a higher number of negative link predictions. For a few datasets (e.g., Cora, CiteSeer, LastFMAsia), a handful of theoretic LP scores are negative because regression (incorrectly) predicts $\overline{\rho}_s^2(b)$ for 1-2 groups $S^{(b)}$ to be negative.

|  | NRMSE ($\downarrow$) | PCC ($\uparrow$) | $\Phi_s$ Test AUC ($\uparrow$) |
|---|---|---|---|
| CORA | $0.034 \pm 0.004$ | $0.830 \pm 0.015$ | $0.915 \pm 0.001$ |
| CITESEER | $0.090 \pm 0.014$ | $0.365 \pm 0.070$ | $0.913 \pm 0.008$ |
| DBLP | $0.026 \pm 0.003$ | $0.652 \pm 0.029$ | $0.933 \pm 0.004$ |
| PUBMED | $0.054 \pm 0.007$ | $0.813 \pm 0.038$ | $0.932 \pm 0.003$ |
| CS | $0.047 \pm 0.008$ | $0.677 \pm 0.036$ | $0.970 \pm 0.001$ |
| PHYSICS | $0.055 \pm 0.007$ | $0.566 \pm 0.026$ | $0.976 \pm 0.001$ |
| LASTFMASIA | $0.049 \pm 0.008$ | $0.682 \pm 0.035$ | $0.960 \pm 0.003$ |
| DE | $0.030 \pm 0.008$ | $0.683 \pm 0.047$ | $0.935 \pm 0.001$ |
| EN | $0.039 \pm 0.006$ | $0.463 \pm 0.022$ | $0.905 \pm 0.002$ |
| FR | $0.031 \pm 0.006$ | $0.654 \pm 0.067$ | $0.935 \pm 0.002$ |

Table 4: **The test AUC of the $\Phi_s$ encoders with an $f_{MLP}$ score function on the real-world network datasets, and the NRMSE and PCC of the theoretic LP scores as predictors of the $\Phi_s$ scores.**

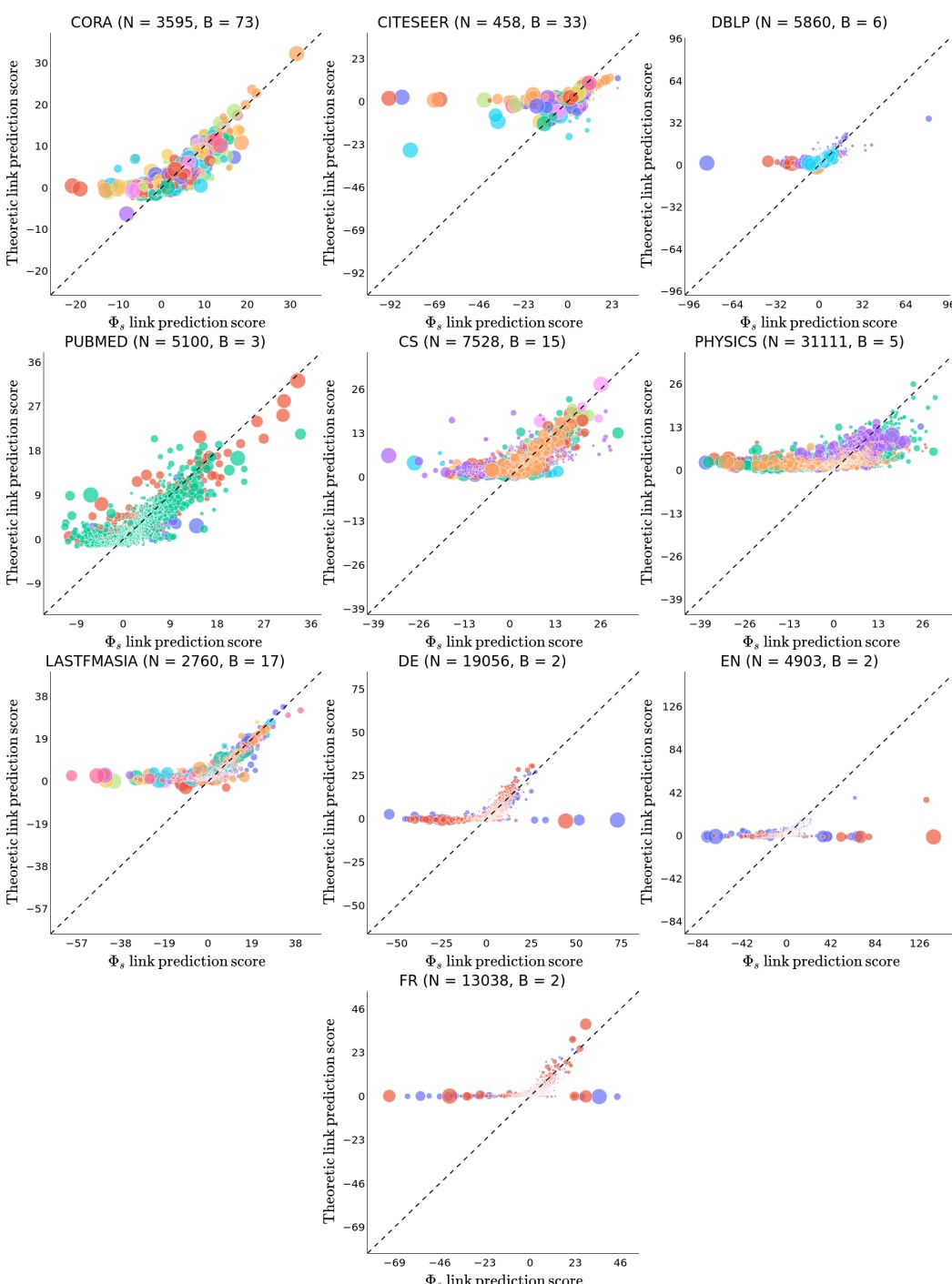

Figure 8: **Theoretic LP score vs. $\Phi_s$ LP score (with Hadamard and MLP) for all network datasets.**

## G.3 Additional experiments for §5.2

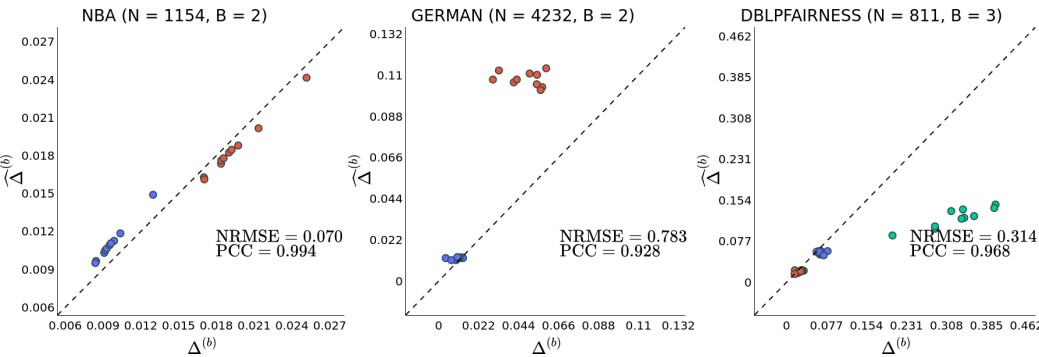

Figure 9: The plots display $\widehat{\Delta}^{(b)}$ vs. $\Delta^{(b)}$ for 4-layer $\Phi_s$ for the NBA, German, and DBLP-Fairness datasets over all $b \in [B]$ and 10 random seeds.

## G.4 Additional experiments for §5.3

| | $\lambda_{\text{fair}}$ | $\mathcal{L}_{\text{fair}}$ ($\downarrow$) | $\Phi_s$ **Test AUC** ($\uparrow$) | | $\lambda_{\text{fair}}$ | $\mathcal{L}_{\text{fair}}$ ($\downarrow$) | $\Phi_r$ **Test AUC** ($\uparrow$) |
|---|---|---|---|---|---|---|---|
| NBA | 4.0 | $0.000 \pm 0.000$ | $0.752 \pm 0.001$ | NBA | 4.0 | $0.000 \pm 0.000$ | $0.754 \pm 0.002$ |
| NBA | 2.0 | $0.006 \pm 0.001$ | $0.752 \pm 0.001$ | NBA | 2.0 | $0.014 \pm 0.003$ | $0.753 \pm 0.000$ |
| NBA | 1.0 | $0.011 \pm 0.001$ | $0.753 \pm 0.001$ | NBA | 1.0 | $0.023 \pm 0.002$ | $0.753 \pm 0.000$ |
| NBA | 0.0 | $0.014 \pm 0.001$ | $0.753 \pm 0.001$ | NBA | 0.0 | $0.029 \pm 0.002$ | $0.753 \pm 0.000$ |
| DBLPFAIRNESS | 4.0 | $0.090 \pm 0.041$ | $0.793 \pm 0.009$ | DBLPFAIRNESS | 4.0 | $0.117 \pm 0.049$ | $0.792 \pm 0.011$ |
| DBLPFAIRNESS | 2.0 | $0.070 \pm 0.015$ | $0.800 \pm 0.007$ | DBLPFAIRNESS | 2.0 | $0.072 \pm 0.013$ | $0.798 \pm 0.006$ |
| DBLPFAIRNESS | 1.0 | $0.099 \pm 0.009$ | $0.804 \pm 0.007$ | DBLPFAIRNESS | 1.0 | $0.111 \pm 0.011$ | $0.797 \pm 0.006$ |
| DBLPFAIRNESS | 0.0 | $0.122 \pm 0.028$ | $0.820 \pm 0.009$ | DBLPFAIRNESS | 0.0 | $0.134 \pm 0.013$ | $0.815 \pm 0.012$ |
| GERMAN | 4.0 | $0.012 \pm 0.008$ | $0.817 \pm 0.004$ | GERMAN | 4.0 | $0.038 \pm 0.007$ | $0.819 \pm 0.006$ |
| GERMAN | 2.0 | $0.018 \pm 0.007$ | $0.827 \pm 0.015$ | GERMAN | 2.0 | $0.042 \pm 0.005$ | $0.835 \pm 0.021$ |
| GERMAN | 1.0 | $0.018 \pm 0.008$ | $0.856 \pm 0.025$ | GERMAN | 1.0 | $0.053 \pm 0.005$ | $0.873 \pm 0.017$ |
| GERMAN | 0.0 | $0.028 \pm 0.007$ | $0.874 \pm 0.011$ | GERMAN | 0.0 | $0.080 \pm 0.008$ | $0.884 \pm 0.008$ |

Table 5: $\mathcal{L}_{\text{fair}}$ and the test AUC for the NBA, German, and DBLP-Fairness datasets with various settings of $\lambda_{\text{fair}}$. The **left** table corresponds to 4-layer $\Phi_s$, and the **right** to 4-layer $\Phi_r$.

# H    THEORY PITFALLS

To understand the second pitfall from §5.1, we separately investigate the association between the degree product $\left(\sqrt{\widehat{\boldsymbol{D}}_{ii}\widehat{\boldsymbol{D}}_{jj}}\right)$ and the absolute deviation of the theoretic LP scores from the $\Phi_s$ scores, and the association between the (transformed) feature similarity $\left(\left\|\sum_{k\in S^{(b)}}\frac{\sqrt{\widehat{\boldsymbol{D}}_{kk}}}{\text{vol}(\mathcal{G}^{(b)})}\alpha_k\right\|_2^2\right)$ and the absolute deviation (cf. Figure 10). We observe that the absolute deviation is highest for the node pairs with a relatively small degree product (i.e., nodes with a low PA score) and low feature similarity.

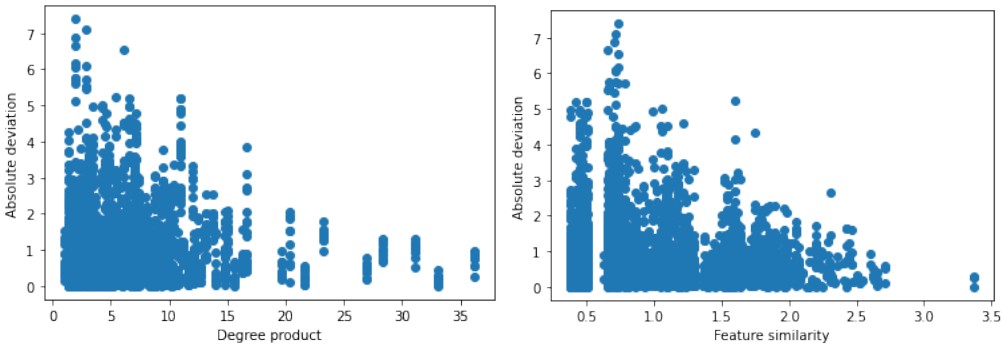

Figure 10: Associations of absolute deviation with degree product and with feature similarity.

## I    ERROR ANALYSIS OF $\Phi_r$ THEORETIC SCORES

**We find in Figure 11 that the relative error (as measured by NRMSE and PCC) of the theoretic link prediction scores for $\Phi_r$ is not lower for lower values of the max term $\max_{u,v \in \mathcal{V}} \sqrt{\frac{\widehat{D}_{vv}}{\widehat{D}_{uu}}}$.**

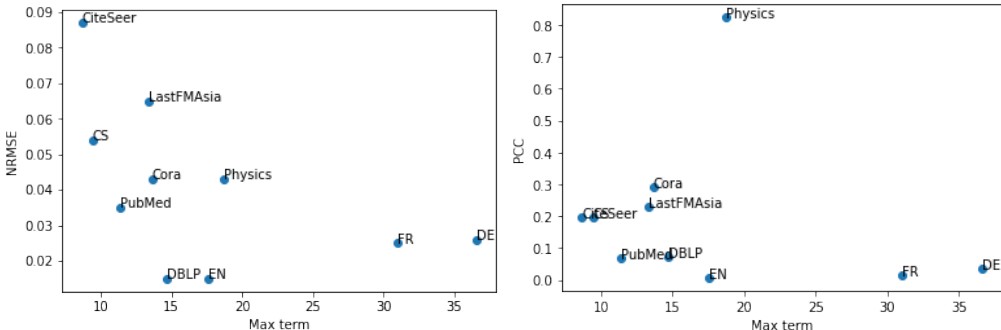

Figure 11: **Weak associations of max term with NRMSE and PCC of theoretic link prediction scores for $\Phi_r$ across all the datasets described in §C.**

**Furthermore, Figure 12 reveals that $\Phi_r$ link prediction scores are *not* higher for incident nodes with larger degrees.**

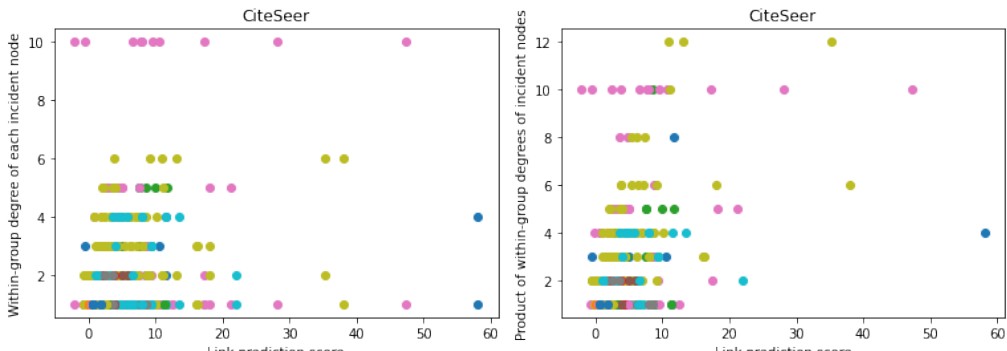

Figure 12: **Weak associations of mean $\Phi_r$ link prediction scores (over 10 random seeds) with degree of each incident node and product of degrees of both incident nodes. Colors correspond to different groups.**

