# OpenReview forum: "Networked Inequality: Preferential Attachment Bias in Graph Neural Network Link Prediction"
_ICLR.cc/2024/Conference — Submitted to ICLR 2024_

### Official Review · Reviewer_LXYw · 2023-10-27

**Soundness:** 3 good
**Presentation:** 3 good
**Contribution:** 2 fair
**Rating:** 5
**Confidence:** 4

**Summary:**

This paper theoretically and empirically shows that graph convolutional networks (GCNs) can exhibit preferential attachment (PA) bias in link prediction, where GCNs tend to predict more links between high-degree nodes that belong to the same social group. The authors propose a simple training-time strategy, based on a fairness regularization term, to mitigate within-group unfairness in GCN link prediction. Experiments show this term reduces unfairness without severely impacting prediction performance.

**Strengths:**

(1) This paper provides abundant theoretical analysis under the specified settings.

(2) This paper proposes a new metric to quantify within-group unfairness in link prediction, which measures disparities in link prediction scores between social groups. The proposed fairness regularizer also provides a simple and effective way to address the newly characterized unfairness.

(3) Experiments are comprehensive in terms of the number of datasets, showing the effectiveness of the proposed training time debiasing solution.

**Weaknesses:**

(1) This paper mainly analyzes the GCN model, failing to consider more widely used alternatives in LP tasks, e.g., SOTA contrastive methods. In addition, this paper relies on relatively simple settings. For example, this paper only considers performing LP with an inner-product decoder, while adopting an MLP classification model on top of the Hadamard product between a pair of node embeddings is more widely used.

(2) There is no baseline adopted for comparison in this paper, and it is not reasonable to avoid such a comparison by claiming the studied problem is novel. It would be necessary to see whether the studied problem is a prevalent problem among different commonly used LP methods.

(3) The evaluation of fairness regularizer utilizes the loss itself as a metric, which seems to be not convincing: the loss would be reduced as long as the gradient descent is effective in most cases.

**Questions:**

(1) Does this studied fairness issue widely exist in those commonly used link prediction models, such as those contrastive GNNs? Is the theoretical analysis in this paper generalizable to them?

(2) Will those commonly used LP models naturally underperform or outperform the reported performance under fairness regularization?

(3) Is there any particular reason why the analysis is performed on GCN? Since the vanilla GCN is not commonly used for LP tasks, this seems questionable to me.

(4) What is the time complexity of the proposed regularization-based method?

---

> ### Author Response · Authors · 2023-11-17
>
> We thank the reviewer for their helpful and specific feedback! To address your stated weaknesses and questions:
>
> - Weakness (1), Questions (1)-(2): By theoretically and empirically uncovering preferential attachment bias and within-group unfairness issues in GCN link prediction, we lay a foundation to study these issues in other link prediction methods in the future. This will include SOTA contrastive methods for link prediction (1) and how they are affected by fairness regularization (2). Overall, this is an important and interesting direction of research.
>
> **We also include experiments for a link predictor that uses a Hadamard product and MLP score function (instead of an inner product) in Appendix G.2. We find that our theoretical analysis is still relevant to and reasonably supports the experimental results, both qualitatively and quantitatively.** This could be because MLPs have an inductive bias towards learning simpler, often linear functions [1, 2], and our theoretical findings are generalizable to linear LP score functions.
>
> - Weakness (2): Because we unveil a new form of unfairness, it is difficult for us to find comparable methods to use as baselines. While we describe methods for alleviating degree bias in the “Degree bias in GNNs” paragraph in our related work section, these methods address performance disparities for low-degree nodes. In contrast, we do not study performance issues but rather how GCN often scales node representations proportionally to the square root of their within-group degree, which affects the magnitude of their link prediction scores.
>
> - Weakness (3): We do not use the loss as our fairness metric, but rather the fairness metric as a regularization term in our loss. We theoretically motivate and ground the fairness metric in Section 4.2, and because the metric is differentiable, we are able to directly add it to our loss function. While it is not surprising that the fairness regularization approach will minimize our metric, it is significant that our experiments show that the regularization does not severely impact link prediction performance.
>
> - Question (3): We focus on GCN (with symmetric and random walk normalized filters) because it provides us with a reasonable setting to develop a rigorous theory of preferential attachment bias in GNN link prediction while leveraging tools from spectral graph theory. We believe that our results may be generalizable to GNN architectures that use a symmetric degree-normalized diffusion operator. Furthermore, it is not uncommon to use vanilla GCN for link prediction; for example, PyTorch Geometric (a popular graph learning library) has code examples that apply vanilla GCN to link prediction: https://github.com/pyg-team/pytorch_geometric/blob/master/examples/link_pred.py.
>
> - Question (4): The time complexity of calculating the regularization term is $O(\sum_{b = 1}^B  |S^{(b)} \cap T^{(1)}| \cdot S^{(b)} +  |S^{(b)} \cap T^{(2)}| \cdot S^{(b)})$, as we have already computed the link prediction scores for the cross-entropy loss term and simply need to sum them appropriately with respect to the groups and subgroups. The time complexity of computing gradients for the regularization term should be on the same order as backpropagation for the cross-entropy loss term.
>
> [1] Nakkiran, Preetum, et al. "Sgd on neural networks learns functions of increasing complexity." arXiv preprint arXiv:1905.11604 (2019).
>
> [2] Valle-Perez, Guillermo, Chico Q. Camargo, and Ard A. Louis. "Deep learning generalizes because the parameter-function map is biased towards simple functions." arXiv preprint arXiv:1805.08522 (2018).

---

> ### Comment · Reviewer_LXYw · 2023-11-23
> **Thanks for the rebuttal**
>
> Thanks for the response from the authors.
>
> My concerns are partially addressed, but two of my major concerns remain: (1) it is not surprising that if a term is added to the objective function, then this term will be optimized when the objective is optimized, so it seems not comprehensive to only present the fairness performance from this specific perspective; (2) There should be a comprehensive discussion about why it is impossible to adopt existing frameworks as baselines for the studied problem. For example, does any of the existing fairness-aware frameworks contribute to the optimization of the proposed fairness metric? How does the proposed fairness metric connect with other existing metrics? It would be more reasonable to avoid adopting any baseline after properly placing the proposed approach in a broader literature background about the studied problem.

---

> > ### Author Response · Authors · 2023-11-23
> >
> > To further address your remaining concerns, we would like to draw your attention to our response to the area chair (which we paste below).
> >
> > > Studies concerning “degree bias” have observed that low-degree nodes experience degraded performance compared to high-degree nodes. They have thus often formulated degree bias from a performance perspective, focusing on equal opportunity. In particular, these studies seek to satisfy $Pr(\hat{y}_v = y | y_v = y, deg(v) = d) = Pr(\hat{y}_v = y | y_v = y, deg(v) = d')$ for all possible degrees $d, d’$, where $\hat{y}_v$ is the prediction for node $v$ and $y_v$ is its ground-truth label. This fairness criterion treats the degree of a node as a sensitive attribute, requiring that a GNN’s accuracy is consistent across nodes with different degrees.
> >
> > > However, we want to ensure that degree disparities in networks are not amplified by GNN link prediction. **We cannot adopt the equal opportunity formulation of degree bias** because it is concerned with performance while we are concerned with degree disparity amplification. For example, even if we consistently predict links with the same accuracy across nodes with different degrees, high-degree nodes can still receive higher link prediction scores than low-degree nodes. In this way, the “degree bias” discussed by other studies is not compatible with our unfairness metric (Eq. 9).
> >
> > > Specifically, we care that $E [ \hat{y}\_{u v} | deg(u) = d ] = E [ \hat{y}\_{u v} | deg(u) = d']$, where $\hat{y}_{u v}$ is the GNN score for a link prediction between nodes $u, v$. In other words, we do not want GNN link prediction scores to be higher for high-degree nodes vs. low-degree nodes. This is what motivates our fairness metric (Eq. 9).
> >
> > > Our theoretical analysis (Eq. 7) and empirical validation reveal that GCN *fundamentally* predicts links between nodes $i, j$ with score $\propto \sqrt{deg(i) \cdot deg(j)}$ *because* of the symmetric normalized filter. This preferential attachment finding allows us to express our unfairness metric in terms of degree disparity (Eq. 10), but this degree disparity is *not* related to the "degree bias" that has been discussed by other papers; this is a new fairness paradigm.

---

### Official Review · Reviewer_bJuY · 2023-10-30

**Soundness:** 3 good
**Presentation:** 4 excellent
**Contribution:** 3 good
**Rating:** 5
**Confidence:** 4

**Summary:**

The paper investigates the bias towards high degree nodes in link prediction when the underlying model is a  Graph Convolutional Network based on one of two filters: the symmetric normalized graph laplacian or on the random walk normalization. It does so by proving two theorems. For the first filter, it shows that the expected raw score output for an edge (i,j) is proportional to the geometric mean of the ("in-block") degrees of the adjacent nodes, under the assumptions of (1) social stratification, (2) expander graphs and that (3) each path from i to j in the computation graph is independently activated with a constant probability dependent on i. For the second filter, it did not uncover a direct relationship with degree. The authors conduct experiments on 10 datasets to validate their theoretical analysis (i.e., comparing the expected raw score with the actual GCN output for several pairs). Moreover, the work bridges this preferential attachment bias and within-group fairness in graph-based recommendation. It proposes a within-group (un-)fairness metric, which measures the disparity among (disjoint) social subgroups within a group. The paper proposes a simple regularization term based on the aforementioned metric to improve fairness and show its efficacy through additional experiments.

**Strengths:**

S1. The paper provides a theoretical analysis to explain preferential attachment biases in GCN-based link prediction.

S2. The assumptions made in the proofs are either supported by experiments or based on empirical evidence from other papers that analyze social network graphs.

S3. Based on the estimate derived for the raw output score, the paper proposes a within-group fairness metric and uses it a (in-processing) fairness regularization method to correct for bias.

S4. The work bridges preferential attachment bias in graph link prediction and current work in within-group fairness.

S5. The text flows nicely and it is very well-written.

**Weaknesses:**

W1. The theoretical analysis is done for two types of filter (symmetric graph Laplacian and random walk), but only the first one is reasonably validated by the experiments. The second one still seems to yield a wide range of scores for different node pairs but this variance is not captured by the estimate. [After a thorough discussion with the reviewers and the AC, I reached the conclusion that this issue needs to be addressed in order to yield a well-rounded paper.]

W2. Some aspects of the initial motivation can be clarified.

W3. There are some other works that attempt to mitigate degree biases in GNN-based link prediction that have not been discussed.

**Questions:**

Q1. The authors offer a potential explanation as to why the theoretic LP scores are not strong predictors of the $\Phi_r$ scores: the extra dependence on the square root of the maximum ration between (in-block) node degrees.
- How to test this conjecture? Did you observe that the relative error is smaller for lower values of this ratio?
- How is the variance in the prediction score related to the node degrees? Did you try plotting a similar graph where the y-axis is some function of $\widehat{D}_i$, $\widehat{D}_j$ or both?
- What are the connections between this result and steady-state of the classic RW on a non-bipartite connected graph?

Q2. Some excerpts were not entirely clear until later in the paper:
- In the explanation for Figure 1, does "social group" refer to gender or discipline?
- In the previous example, if men may receive more collaboration recommendations, why not to fix the maximum number of recommendations per individual? Fewer recommendations could be provided if the model is not very confident about some of them. Is it a problem of calibration (i.e., the model tends to make overconfident predictions for certain subgroups)?
- In  Eq. (5), which ones takes precedence: exponentiation to the L-th power or subscripting ij? Consider using
$[(D^{-1/2} A D^{-1/2})^L]_{ij}$ and $[( D^{-1} A)^L]_{ij}$.

Q3. Are you familiar with these works? Please discuss whether they should be included as part of related work.
- Kojaku, Sadamori, Jisung Yoon, Isabel Constantino, and Yong-Yeol Ahn. "Residual2Vec: Debiasing graph embedding with random graphs." Advances in Neural Information Processing Systems 34 (2021): 24150-24163.
- Harry Shomer, Wei Jin, Wentao Wang, and Jiliang Tang. 2023. Toward Degree Bias in Embedding-Based Knowledge Graph Completion. In Proceedings of the ACM Web Conference 2023 (WWW '23). Association for Computing Machinery, New York, NY, USA, 705–715. https://doi.org/10.1145/3543507.3583544

---

> ### Author Response · Authors · 2023-11-17
>
> We thank the reviewer for their detailed and valuable feedback! To address your concerns and questions:
>
> - W2: Could you please elaborate on which aspects of the initial motivation can be clarified? (e.g., specific lines). We greatly appreciate your help in making our motivation clear and accessible to readers.
>
> - W3, Q3: Thank you for bringing [1, 2] to our attention; we will add these papers to the “Degree bias in GNNs” paragraph in our related work section. However, these works touch upon different forms of degree bias (i.e., sampling and performance bias) than the type of degree bias we uncover: GCNs often scale node representations proportionally to the square root of their within-group degree, which affects the magnitude of their link prediction scores.
>
> - Q1: **We did not observe that the relative error is smaller for lower values of the ratio. We add plots demonstrating this in Appendix I (Figure 11).**
>
> **Furthermore, $\Phi_r$ link prediction scores are not associated with either $\widehat{D}_{i i}$, $\widehat{D}_{j j}$, or the product thereof. We add plots demonstrating this in Section I (Figure 12).**
>
> There are intimate connections between our result and the steady-state probabilities of the classic RW. The stationary probabilities of classic RW are the same regardless of the starting node. This is why $\Phi_r$ produces similar representations for all the nodes in each group, regardless of the degree of the node (with a larger number of layers, $\Phi_r$ would oversmooth all the representations to the same vector). Hence, $\Phi_r$ link prediction scores do not have a degree dependence, theoretically or empirically.
>
> - Q2: In Figure 1, we treat the disciplines as groups and gender as subgroups; in particular, we are concerned about GCN link prediction further marginalizing the subgroup of women within the group of CS researchers.
>
> We agree that fixing the number of recommendations per individual is a possible solution, but doing this with utility in mind requires identifying a utility-maximizing subset of link predictions. As our theoretical and empirical results reveal, GCN link prediction scores are often inherently proportional to the geometric mean of the degrees of the incident nodes, which can make them a poor indicator of prediction confidence. From a calibration perspective, GCN naturally makes overconfident predictions for links between high-degree nodes.
>
> In Eq. (5), the power takes precedence over the subscripting.
>
> [1] Kojaku, Sadamori, Jisung Yoon, Isabel Constantino, and Yong-Yeol Ahn. "Residual2Vec: Debiasing graph embedding with random graphs." Advances in Neural Information Processing Systems 34 (2021): 24150-24163.
>
> [2] Harry Shomer, Wei Jin, Wentao Wang, and Jiliang Tang. 2023. Toward Degree Bias in Embedding-Based Knowledge Graph Completion. In Proceedings of the ACM Web Conference 2023 (WWW '23). Association for Computing Machinery, New York, NY, USA, 705–715. https://doi.org/10.1145/3543507.3583544

---

> > ### Comment · Reviewer_bJuY · 2023-11-17
> >
> > I appreciate the authors' thoughtful responses and additional experiments to improve the paper.
> >
> > > W2: Could you please elaborate on which aspects of the initial motivation can be clarified? (e.g., specific lines).
> >
> > Those aspects were specified in Q2. Thank you for addressing them.

---

### Official Review · Reviewer_LDLM · 2023-10-30

**Soundness:** 3 good
**Presentation:** 3 good
**Contribution:** 3 good
**Rating:** 8
**Confidence:** 3

**Summary:**

This paper explores the impact of degree bias in networks on Graph Convolutional Network (GCN) link prediction (LP). The authors investigate how the preferential attachment mechanism, which creates degree discrepancies between nodes, can impact link prediction scores. Moreover, they explore within-group fairness to investigate if the bias in link prediction is additionally enlarged by considering subgroups considering two attributes, such as ethnic background and gender. The research focuses on GCNs with symmetric and random walk normalized graph filters and examines their LP scores within the same social group. They find that GCNs with symmetric normalized filters exhibit within-group preferential attachment bias in link prediction. This bias can result in disparities in link prediction scores between social groups, potentially amplifying degree and power imbalances in networks.

In particular, the authors provide a theoretical analysis of a within-group preferential attachment bias in link prediction of GCNs with symmetric normalized graph filters. They empirically validate these findings on 10 real-world networks. For GCNs with a random walk normalized filter, the authors theoretically do not find a PA bias, which is however contradicted by empirical evidence. Building on these findings, the authors contribute a new within-group fairness metric for LP, which quantifies disparities in LP scores between social groups. Lastly, the authors propose a training-time strategy to alleviate within-group unfairness, which they assess on three real-world networks revealing its effectiveness.

**Strengths:**

The manuscript is well written and, in my opinion, a valuable contribution to the literature. The authors carefully derive their theoretical results and perform experiments on real-world datasets which (mostly) back up their results. Even when the authors find discrepancies between their theoretical predictions and their experiments, the limitations are discussed appropriately.
The extension of the fairness assessment to within-group fairness is an important consideration that is often missing in the current literature. The authors also point towards intersectionality literature, which would be an interesting extension which is probably not possible due to space constraints.

**Weaknesses:**

Right before Section 4.2 the authors rightly state that “such “rich get richer” dynamics can engender group unfairness when nodes’ degrees are statistically associated with their group membership…”

Whether or whether not nodes’ degrees are statistically associated with their group membership largely depends on their group size and homophily of the interactions. Maybe a discussion of the impact of homophily would be appropriate here.
The authors only state in their future work, that it would be useful to study heterophilic networks as well, but never touch on the concept of homophily in the rest of the paper.

Moreover, it would have been interesting if there would have been a larger discussion of the interpretation of the different intersections of groups in the within-fairness part and how marginalisation of certain social groups paper aligns or contradicts with social science literature.

In Figure 2, a legend of the colour code of the dots would be helpful.

To me, the adaption of node classification datasets to LP did not become as clear. Is it true, that the labels are associated to network structure and are now used as the group truth for the groups? If this is true, the networks would anyways be largely homophilic, that could be stated somewhere.

Very minor: page 21 C: “We we row-normalise…”

**Questions:**

see weaknesses

---

> ### Author Response · Authors · 2023-11-17
>
> We thank the reviewer for their insightful and thorough feedback! To address your concerns:
>
> - We agree that the association between node degrees and group membership depends on group size and homophily. In our camera-ready (after working out space constraints), we will add the following text before Section 4.2:
>
> *”An association between node degrees and group membership depends on group size and homophily; in particular, when a group has many nodes and intra-links (i.e., is homophilous), there may be more nodes with a high within-group degree.”*
>
> We touch upon social group homophily in Section 4.1 using the term “social stratification” from the social science literature. In particular, the intra-link approximation error term $\sum_{l = 1}^L {L \choose l} \left\| \Xi^{(0)} \right\|^l_{op} \left\| \widehat{P} \right\|^{L - l}_{op}$ captures homophily.
>
> - We will add a discussion of how our findings relate to Intersectionality in the “Within-group fairness” paragraph of Section 2:
>
> *”Our theoretical and empirical findings reveal that GCN link prediction can further marginalize social subgroups. This relates to the “complexity” tenet of Intersectionality, which expresses that the marginalization faced by, e.g., Black women, is non-additive and distinct from the marginalization faced by Black men and white women [1].”*
>
> - In Figure 2, the colors represent different groups. We describe the type of groups in each dataset in Section C; for example, in the LastFMAsia dataset, the groups are the home countries of users. However, we were unable to find the exact group names (e.g., Vietnam, India) from the dataset documentation.
>
> - We adopt the class labels for each dataset as the social group labels. This design choice is reasonable, as in all the datasets, the classes naturally correspond to socially-relevant groupings of the nodes, or proxies thereof (e.g., in the LastFMAsia dataset, the classes are the home countries of users). We will add the following discussion of homophily in Section 5:
>
> *”Because we adopt the class labels for each dataset as the social group labels, the social groups are largely homophilic. This aligns with our assumptions when interpreting Theorems 4.3 and 4.4 that social groups are stratified in networks.”*
>
> [1] Collins, Patricia Hill, and Sirma Bilge. Intersectionality. John Wiley & Sons, 2020.

---

> > ### Comment · Reviewer_LDLM · 2023-11-18
> > **thanks for the answer**
> >
> > I have increased my score accordingly.

---

### Official Review · Reviewer_qGCf · 2023-11-02

**Soundness:** 3 good
**Presentation:** 4 excellent
**Contribution:** 3 good
**Rating:** 6
**Confidence:** 3

**Summary:**

This paper studies the fairness of link prediction in Graph Neural Networks (GNN), focusing on within-group fairness and the "rich get richer" effect in networks. Its main result, as given in Theorem 4.3, is that GCNs with symmetric normalized graph filters exhibit a bias toward within-group preferential attachment. Numerical experiment verifies this theoretical result to a good extent.

**Strengths:**

1. The paper has good mathematical rigor, and the results are significant.
2. I find Lemma 4.1 and Theorem 4.3 interesting and enjoyable to read. They should be of great interest to community interested in deciphering the societal implications of GNN-based LP when they are deployed on large-scale social systems.
3. The experiments are well-designed and well support the theory.

**Weaknesses:**

1. The assumption about the independence of path activation probabilities (ρs(i) and ρr(i)) is rather strong and may not hold true in real world. This can have great effect on the theoretical result. It would be helpful to discuss more.

2. Canonical GNNs nowadays are rarely used for link prediction task due to some of their inherent limitation. Some of the classical works on link predictions, like [1, 2, 3], all use some additional signals one top canonical GNNs. It would be great, if possible, to also give some theoretical discussions on these works.




[1] Graph Neural Networks for Link Prediction with Subgraph Sketching, ICLR 2023
[2] Link Prediction Based on Graph Neural Networks, NeurIPS 2018
[3] Distance Encoding: Design Provably More Powerful Neural Networks for Graph Representation Learning, NeurIPS 2020

**Questions:**

See weaknesses.

---

> ### Author Response · Authors · 2023-11-17
>
> We thank the reviewer for their thoughtful feedback! To address your concerns:
>
> - We agree that the independence of path activation probabilities may not always hold true in the real world. However, we verify that this assumption is plausible via our extensive experiments on real-world datasets to validate our theoretical analysis (Section 5.1). This assumption also aligns with findings that deep neural networks have an inductive bias towards learning simpler, often linear functions [1, 2]. Furthermore, a variant of our assumption (where $\rho(i) = \rho$ is constant for all nodes) has been used in the literature to simplify theoretical analysis (e.g., [3, 4]); our assumption may be more realistic than this variant, as it captures that the probability of paths activating can differ across nodes (e.g., due to differences in features, neighborhood structure).
>
> - Thanks for highlighting works [5, 6, 7] on link prediction using GNNs. To address these, we will add the following text to the end of Section 3 in our camera-ready (once we work out space constraints):
>
> *"In recent years, researchers have proposed methods to improve the expressivity of node representations for link prediction by capturing subgraph information [5, 6, 7]. Our theoretical findings remain relevant to methods that ultimately use a GCN to predict links (e.g., [6, 7]), as we do not make assumptions about the features passed to the GCN (i.e., they could be distance encodings, SEAL node embeddings, etc.) Studying the fairness of more expressive link prediction methods is an interesting direction for future research."*
>
> **We also include experiments for a link predictor that uses a Hadamard product and MLP score function (instead of an inner product) in Appendix G.2. We find that our theoretical analysis is still relevant to and reasonably supports the experimental results, both qualitatively and quantitatively.** This could be because MLPs have an inductive bias towards learning simpler, often linear functions [1, 2], and our theoretical findings are generalizable to linear LP score functions.
>
> [1] Nakkiran, Preetum, et al. "Sgd on neural networks learns functions of increasing complexity." arXiv preprint arXiv:1905.11604 (2019).
>
> [2] Valle-Perez, Guillermo, Chico Q. Camargo, and Ard A. Louis. "Deep learning generalizes because the parameter-function map is biased towards simple functions." arXiv preprint arXiv:1805.08522 (2018).
>
> [3] Xu, Keyulu, et al. "Representation learning on graphs with jumping knowledge networks." International conference on machine learning. PMLR, 2018.
>
> [4] Tang, Xianfeng, et al. "Investigating and mitigating degree-related biases in graph convoltuional networks." Proceedings of the 29th ACM International Conference on Information & Knowledge Management. 2020.
>
> [5] Graph Neural Networks for Link Prediction with Subgraph Sketching, ICLR 2023
>
> [6] Link Prediction Based on Graph Neural Networks, NeurIPS 2018
>
> [7] Distance Encoding: Design Provably More Powerful Neural Networks for Graph Representation Learning, NeurIPS 2020

---

> > ### Comment · Reviewer_qGCf · 2023-11-22
> >
> > Thanks for the response. My concerns have been resolved, and I would like to maintain my current score.

---

### Author Response · Authors · 2023-11-17

We thank all the reviewers for their positive feedback and thoughtful comments on our work.

- Reviewer qGCf finds that the theoretical contributions of the paper have “good mathematical rigor” and are “significant” and “interesting,” as well as that our experiments are “well-designed.”

- Reviewer LDLM states that the paper is “well written” and “a valuable contribution to the literature.”

- Reviewer bJuY concurs that the manuscript “flows nicely” and is “very well-written.”

- Reviewer LXYw adds that the paper provides “abundant theoretical analysis,” “a simple and effective way to address the newly characterized unfairness,” and comprehensive experiments.

We address each reviewer’s concerns and questions individually. We made updates to the paper’s appendix in a magenta font color, and we propose additional updates in our individual responses to reviewers (that we will add to our camera-ready, once we work out space constraints).

---

> ### Comment · Area_Chair_1niT · 2023-11-19
> **Questions from AC**
>
> Dear Authors,
>
> I am checking your paper and have several questions, which can hopefully get your further clarification.
>
> **Motivation** I read the abstract, introduction, and related work several times, but failed to find a concrete motivation. Based on my understanding, the motivation comes from the challenges of the research question and drawbacks of existing solutions. I am eager to know which challenges this paper targets to solve and how difficult they are, and why the existing solutions cannot be adapted to solve these challenges, or why the targeted gap is important. Here I am asking for the motivation, rather than the research question or solutions. Please do not repeat some sentences like the last part of the first paragraph in related work, which does not help with this concern.
>
> **PA** What is preferential attachment? Could you provide a formal definition or equation? Further, what is the new type of unfairness in this paper? Any formal definition or equation?
>
> **Metric** What metrics are used for evaluating the unfairness?
>
> Look forward to hearing from the authors. Thank you.
>
> Kind Regards,
>
> AC

---

> ### Author Response · Authors · 2023-11-21
>
> We thank the area chair for their questions and are happy to provide clarification. We begin by explaining preferential attachment, before elaborating on our motivation and how we define unfairness.
>
> **PA:** Preferential attachment describes the propensity of links to form with high-degree nodes (https://networkx.org/documentation/stable/reference/algorithms/generated/networkx.algorithms.link_prediction.preferential_attachment.html). Network scientists have studied for decades how links in real-world networks exhibit preferential attachment. For example, in the iterative Barabasi-Albert model of network formation, each new node $s$ forms links with existing nodes $t$ with probability proportional to the degree of $t$, i.e., $Pr((s, t) \in {\cal E}) \propto deg(t)$. In the context of our paper, preferential attachment describes how a GCN (with an inner product score function) often predicts links between nodes $i, j$ with score $\propto \sqrt{deg(i) \cdot deg(j)}$ (Eq. 7).
>
> **Motivation:** A wealth of literature in network science and the social sciences has examined the preferential attachment properties of real-world networks and how these properties contribute to unfair (non-neural) algorithms. For example, [1] finds that Instagram accounts run by men have a significantly higher following than those run by women due to gender discrimination; this degree disparity is only amplified by link recommendation algorithms that suggest accounts to follow (i.e., recommending accounts with higher degree to follow, which makes the rich get richer), revealing that these algorithms have a preferential attachment bias. Moreover, many papers outside graph learning have discussed the intersectional unfairness of machine learning.
>
> However, despite the increasing real-world deployment of graph neural networks (GNNs) for link prediction, their unfairness has not been studied from the perspectives of preferential attachment and intersections of social groups. Our paper fills this gap by providing thorough theoretical and empirical evidence that GCNs (an extremely popular GNN architecture with 28843 citations) have a preferential attachment bias when predicting links between nodes in the same social group. **This finding is nontrivial and as GCNs leverage a combination of features and local structural context to make link predictions.**
>
> This research question is challenging from a technical perspective, as it requires uncovering properties of *short* random walks on graphs (since most GNNs are shallow) with social aspects in mind; in contrast, most random walk results in the literature concern random walks at convergence. This is further an important research question because GNNs with a preferential attachment bias can amplify degree disparities, which translates to increased discrimination and disparities in social influence among nodes.
>
> As we uncover this new form of unfairness, there are no existing solutions to this unfairness in the literature. We propose a training-time regularization-based fairness method that alleviates this unfairness without greatly sacrificing the test AUC of link prediction. While we describe methods for alleviating degree bias in the “Degree bias in GNNs” paragraph in our Related Work section, **these methods address degraded performance for low-degree nodes, not preferential attachment bias.** We do not study performance issues but rather how GCN scales node representations proportionally to the square root of their within-group degree, which affects the magnitude of their link prediction scores.
>
> In summary, we augment the field’s understanding of degree biases beyond performance disparities across nodes. We further lay a foundation to study preferential attachment biases and within-group unfairness in GNN link prediction, which is a critical direction of research.
>
> **Unfairness and Metric:** We formally define our new type of unfairness via our unfairness metric in Eq. 9. This metric quantifies disparities in how a GNN allocates link prediction scores between social subgroups (i.e., are links with nodes in one subgroup predicted at a higher rate than links with nodes in the other subgroup?). Our metric is motivated by how GNN link predictions (e.g., in recommender systems) influence real-world link formation, which has consequences for degree and power disparities.
>
> We use Eq. 9 to evaluate unfairness. We cannot simply adopt common fairness metrics like node-wise equal opportunity or dyadic fairness, as they do not capture the new type of unfairness that we uncover.
>
> Please let us know if we can answer any additional questions.
>
> [1] Stoica, Ana-Andreea, Christopher Riederer, and Augustin Chaintreau. "Algorithmic glass ceiling in social networks: The effects of social recommendations on network diversity." Proceedings of the 2018 World Wide Web Conference. 2018.

---

> > ### Comment · Area_Chair_1niT · 2023-11-21
> > **Response to Authors**
> >
> > Dear Authors,
> >
> > Thanks for your response. I am still trying to understand the new fairness proposed in this paper. What is the difference between degree bias and preferential attachment bias?
> >
> > Based on my knowledge, there are tons of studies that set the node degree as a sensitive attribute and propose some new algorithms to tackle the degree bias. In this paper, the authors focus on preferential attachment bias, where each edge $e_{ij}$ can be set a preferential attachment score with $deg(i)\cdot deg(j)$, and the authors target this new bias for link prediction. Do I have the right understanding?
> >
> > Kind Regards,
> >
> > AC

---

> > > ### Author Response · Authors · 2023-11-21
> > >
> > > Studies concerning “degree bias” have observed that low-degree nodes experience degraded performance compared to high-degree nodes. They have thus often formulated degree bias from a performance perspective, focusing on equal opportunity. In particular, these studies seek to satisfy $Pr(\hat{y}_v = y | y_v = y, deg(v) = d) = Pr(\hat{y}_v = y | y_v = y, deg(v) = d')$ for all possible degrees $d, d’$, where $\hat{y}_v$ is the prediction for node $v$ and $y_v$ is its ground-truth label. This fairness criterion treats the degree of a node as a sensitive attribute, requiring that a GNN’s accuracy is consistent across nodes with different degrees.
> > >
> > > However, we want to ensure that degree disparities in networks are not amplified by GNN link prediction. **We cannot adopt the equal opportunity formulation of degree bias** because it is concerned with performance while we are concerned with degree disparity amplification. For example, even if we consistently predict links with the same accuracy across nodes with different degrees, high-degree nodes can still receive higher link prediction scores than low-degree nodes. In this way, the “degree bias” discussed by other studies is not compatible with our unfairness metric (Eq. 9).
> > >
> > > Specifically, we care that $E [ \hat{y}\_{u v} | deg(u) = d ] = E [ \hat{y}\_{u v} | deg(u) = d']$, where $\hat{y}_{u v}$ is the GNN score for a link prediction between nodes $u, v$. In other words, we do not want GNN link prediction scores to be higher for high-degree nodes vs. low-degree nodes. This is what motivates our fairness metric (Eq. 9).
> > >
> > > Our theoretical analysis (Eq. 7) and empirical validation reveal that GCN *fundamentally* predicts links between nodes $i, j$ with score $\propto \sqrt{deg(i) \cdot deg(j)}$ *because* of the symmetric normalized filter. This preferential attachment finding allows us to express our unfairness metric in terms of degree disparity (Eq. 10), but this degree disparity is *not* related to the "degree bias" that has been discussed by other papers; this is a new fairness paradigm.

---

> > > > ### Comment · Area_Chair_1niT · 2023-11-22
> > > > **Response to Authors**
> > > >
> > > > Thanks for the further explanations. I will check the paper again and discuss with reviewers.
> > > >
> > > > Kind Regards,
> > > >
> > > > AC

---

### Meta-Review · Area_Chair_1niT · 2023-12-04

**Metareview:**

I have read all the materials of this paper including the manuscript, appendix, comments, and response. Based on collected information from all reviewers and my personal judgment, I can make the recommendation on this paper, *reject*. No objection from reviewers who participated in the internal discussion was raised against the reject recommendation.

**Presentation**

I usually put the comments on presentation in the last point. However, the presentation of this paper heavily affects my understanding of this paper, which makes me have to directly communicate with the authors. Fortunately, with the help from the authors and after my reading of this paper more than 5 times, I figured out what this paper talks about. Note that this paper is just within my expertise area.

1. "GNN LP" in the title seems weird to me. Is "GNN-based LP" better?

2. The research question is not clearly illustrated in the introduction part. Moreover, there is no motivations in the introduction part, either. I have to dig the motivations piece by piece in the related work section.

3. There is no formal definition of the fairness targeted in this paper. In addition, there is no corresponding metric, either.

4. The meaning of colors in Figure 2 should be illustrated in the legend or in the caption. I have no idea why the authors put two tables in Figure 2. It is better to put these two tables outside.

Minor issues that can be easily fixed within one week, are not the deterministic factors. I just want to provide more comments for authors' next-round polish.

**Research Question**

After communicating with the authors, I understand the fairness addressed in this paper is the preferential attachment and intersections of social groups. I did not see a word like "intersection" in the introduction part. I only saw some words like "two social axes" in the third paragraph of related work. What is the benefit to consider two social axes?

**Motivations & Challenge Analysis**

Again, I did not find a paragraph of motivations. Instead, I dug the motivations piece by piece in the related work section. From these sentences, the authors emphasized the difference between their research question and literature. Unfortunately, it is not enough. Beyond the point that no one has done this before, the authors should illustrate how difficult the targeted research question it is; in another word, why the existing methods cannot be adapted to solve the targeted research question.

**Philosophy**

The authors provided some theoretical analyses on GCN with a symmetric normalized graph filter and a random walk normalized filter for LP. These analyses lead to a fairness regularizer in Section 4.4.

This theorem-guided algorithm design is a plus. Its effectiveness depends on whether the practices meet the assumptions of theorems.

**Theoretical Analyses**

Respectfully, the theoretical results are incremental to me, which can be derived from the GNN theoretical systems without too much efforts.

**Techniques**

The authors proposed a new fairness regularizer.

**Experiments**

1. The empirical results conflict with the theoretical analysis on a random walk normalized filter. The correctness or assumption of these theorems are questionable.

2. No competitive methods. I believe there is no work addressing the same research question. However, the existing studies can be easily extended to consider two social axes.

3. No evaluation metric for the fairness proposed in this paper. In Table 1, the authors use the loss of fairness as a metric; this is prohibitive. The evaluation metric should come from the ground truth, which should be algorithm-invariant.

**Justification For Why Not Higher Score:**

This paper is not self-standing and does not reach the bar of ICLR.

**Justification For Why Not Lower Score:**

N/A

---

### Decision · Program_Chairs · 2024-01-16

Reject